# Perturbed Wnt signaling leads to neuronal migration delay, altered interhemispheric connections and impaired social behavior

Riccardo Bocchi[1], Kristof Egervari[1], Laura Carol-Perdiguer [1], Beatrice Viale[1], Charles Quairiaux[1], Mathias De Roo[1], Michael Boitard[1], Suzanne Oskouie[1], Patrick Salmon[1] & Jozsef Z. Kiss[1]

Perturbed neuronal migration and circuit development have been implicated in the pathogenesis of neurodevelopmental diseases; however, the direct steps linking these developmental errors to behavior alterations remain unknown. Here we demonstrate that Wnt/C-Kit signaling is a key regulator of glia-guided radial migration in rat somatosensory cortex. Transient downregulation of Wnt signaling in migrating, callosal projection neurons results in delayed positioning in layer 2/3. Delayed neurons display reduced neuronal activity with impaired afferent connectivity causing permanent deficit in callosal projections. Animals with these defects exhibit altered somatosensory function with reduced social interactions and repetitive movements. Restoring normal migration by overexpressing the Wnt-downstream effector C-Kit or selective chemogenetic activation of callosal projection neurons during a critical postnatal period prevents abnormal interhemispheric connections as well as behavioral alterations. Our findings identify a link between defective canonical Wnt signaling, delayed neuronal migration, deficient interhemispheric connectivity and abnormal social behavior analogous to autistic characteristics in humans.

---

[1] Department of Basic Neurosciences, University of Geneva Medical School, CH-1211 Geneva 4, Switzerland. Riccardo Bocchi and Kristof Egervari contributed equally to this work.  Correspondence and requests for materials should be addressed to J.Z.K. (email: Jozsef.Kiss@unige.ch)

Cortical pyramidal neurons are generated in the ventricular/ subventricular zone of the dorsal pallium from where they migrate along radial glia processes to reach their final position in the cortical plate[1]. Radial migration of neurons is a complex multistep process involving an initial phase of multipolar migration followed by multipolar-to-bipolar transition and glia-guided locomotion[2, 3]. These steps are precisely orchestrated by genetically programed signaling sequences ensuring that neurons arrive in the right time and the right place in specific cortical layers[4]. Migration deficits could result in permanently displaced (i.e., heterotopic) neurons and severe layer malformations leading to epilepsy, psychiatric disorders, cerebral palsy and mental retardation[5]. It is generally presumed that the timing of migration and positioning could also be crucial for the proper integration of neurons into cortical circuits[6]. While the structural and functional consequences of a permanent disruption of neuronal migration are well documented[7], little is known about how transient migration delays without apparent structural defects may impact on the subsequent development of cortical circuits. This is an important issue since it has been suggested that neuronal migration disorders may play a role in the pathogenesis of perturbed neuronal connectivity underlying neuropsychiatric diseases, including autism spectrum disorders (ASD) and schizophrenia[8–12]. Nonetheless, neuroimaging and neuropathological examinations often show an absence of overt cortical malformations in these patients, though mild cortical heterotopias may occur[11].

Wnt signaling is critical in early cortical development[13]; it directs neurogenesis and neuronal differentiation[14, 15]. In addition, non-canonical Wnt signaling pathways have been implicated in circuit assembly, in particular in dendritic development and synaptogenesis of pyramidal neurons in culture[16]. Little is known about the functions of Wnt signaling pathways in regulating radial migration of pyramidal neurons. Canonical Wnt/β-catenin signaling has been reported to play a role in switching neuronal progenitors in the ventricular/subventricular zone from a proliferative mode to a migration mode[17]. In a previous study, we showed that dynamically regulated activity levels of Wnt/β-catenin as well as non-canonical Wnt signaling are essential in the control of multipolar-to-bipolar transition of layer 2/3 pyramidal neurons during the initial phase of radial migration[18]. Here we discovered that canonical Wnt/β-catenin signaling plays a critical role in regulating the timing of radial glia-guided locomotion of callosal projection neurons (CPNs). Reversible inhibition of Wnt/β-catenin signaling induced a transient delay during radial migration of CPNs in the rat somatosensory cortex via downregulating C-Kit expression and, consequently, C-Kit-mediated neuro-glial attachment. Late arrival of migrating neurons initiated a cascade of developmental malformations starting with reduced short-range connectivity and critically low CPN activity, leading to severe abnormalities of homotypic interhemispheric projections. Adult rats with delayed neurons not only showed defective somatosensory function, but developed repetitive movements and impairment in social interactions as well. Restoring normal migration speed via C-Kit overexpression prevented these circuit as well as behavioral alterations, thus supporting a link between canonical Wnt signaling, neuronal migration, perturbed connectivity, and behavioral changes reminiscent of those seen in humans with autism. Importantly, postnatal chemogenetic induction of neuronal activity was also able to rescue interhemispheric projections and social behavior, hence raising the possibility of correcting circuit alterations during the postnatal period.

## Results

### Transient downregulation of Wnt signaling delays migration.
To explore the role of Wnt canonical signaling in glia-guided

locomotion of pyramidal neurons, we first examined signaling activity in migrating neurons using TOPdGFP, a previously described[18] and validated (Supplementary Fig. 1a) Wnt canonical signaling-dependent reporter construct. We introduced this reporter into precursors of layer 2/3 pyramidal neurons of the somatosensory cortex using intrauterine electroporation at embryonic day (E) 18. Single-cell confocal time-lapse imaging revealed that canonical Wnt signaling is dynamically active in neurons during glia-guided locomotion (Fig. 1a and Supplementary Movie 1), and increases as cells approach the pial surface (Fig. 1b). The observed Wnt signaling activity distribution is consistent with the expression pattern of known Wnt canonical ligands (Supplementary Fig. 1b and ref. [18]). To investigate how the dynamic states of Wnt signaling affect CPN migration, we carried out Wnt loss-of-function (LOF) experiments by overexpressing a well-established dominant negative form of Wnt-effector transcription factor (dnTCF4) in L2/3 neurons[19]. The dnTCF4 construct binds DNA at a specific Wnt responsive element but lacks β-catenin binding site, therefore represses transcription of canonical Wnt signaling target genes. Cells expressing the dnTCF4 plasmid showed a significantly lower level of Wnt signaling activity as detected by in vitro luciferase assays (Supplementary Fig. 1c) and in vivo TOPdGFP/dnTCF4 co-electroporation experiments (Supplementary Fig. 1a). To exclude transcriptional LOF effects on neuronal development before and after migration, we induced dnTCF4 expression between E21 and postnatal day (P) 3 (i.e., specifically during migration). The electroporations involved about 30–40% of layer 2/3 pyramidal cells and elicited a significant reduction in migratory speed (Fig. 1c and Supplementary Movie 2). On the other hand, dnTCF4 overexpression did not seem to affect the radial glia structure (Supplementary Fig. 1d). The emergence of delayed neurons in the intermediate zone and in the cortical plate during the late phase of migration (Fig. 1d), however, did not appear to be permanent. By P7, nearly all delayed neurons arrived at their correct laminar level (Fig. 1e, f and Supplementary Fig. 1e). Moreover, these neurons preserved fate-specific marker expression (Supplementary Fig. 1f) and later they exhibited normal dendritic development (Supplementary Fig. 1b, g). To further confirm the role of Wnt signaling in glia-guided locomotion, we targeted an intermediate element of signaling by overexpressing an inducible, dominant negative form of dishevelled-2 (ΔDVL2), which lacks the DIX subdomain. The DIX domain of the dishevelled protein is essential for canonical Wnt signaling[20] and cells expressing this transgene showed a decreased canonical Wnt signaling activity as measured by in vitro luciferase assays (Supplementary Fig. 1c). Similar to the experiments with dnTCF4, we induced expression of ΔDVL2 between E21 and postnatal day P3, and we observed that ΔDVL2-expressing cells displayed a significantly reduced migratory speed (Fig. 1c). The reduction of migratory speed resulted in a delayed neuronal migration at P3 (Fig. 1d). Since the canonical Wnt ligand Wnt3A is expressed in the developing cortex during the period of radial migration (Supplementary Fig. 1b and ref. [18]), we investigated if it could be involved and act cell-autonomously in radial migration. We observed that knockdown of Wnt3A by overexpressing an inducible shRNA (shWNT3A; Supplementary Fig. 1i) resulted in delayed radial migration, although a lesser degree compared to the effect of dnTCF4 (Supplementary Fig. 1j). Thus, WNT3A expressed by CPNs could contribute to the regulation glia-guided locomotion cell autonomously.

Taken together, we conclude that Wnt canonical signaling during glia-guided migration regulates the migration speed of CPNs and therefore the time of their arrival to layer 2/3.

**Wnt signaling regulates attachment to radial glia via C-Kit**. To gain a mechanistic insight into the migration deficit, we examined the effects of dnTCF4 overexpression on the morphology of migrating cells. We did not find any major morphological changes during migration in the intermediate zone. In the cortical plate, however, single-cell time-lapse imaging revealed significantly shorter leading process of dnTCF4 cells compared to that of controls (Fig. 2a, b and Supplementary Movie 3). We also observed that Wnt-signaling-deficient neurons often displayed an inversion of the leading process from the dorsal (i.e., towards pial surface) to ventral (i.e., towards ventricle) direction, which made cells taking pauses more frequently during locomotion (Fig. 2c and Supplementary Movie 4). The change in cellular orientation has been confirmed by post hoc quantitative analysis based on the localization of the Golgi (Supplementary Fig. 2). In particular, the percentage of cells oriented towards the pial surface in the CP dramatically decreased upon dnTCF4 overexpression (Supplementary Fig. 2a). Furthermore, the analysis of orientation angle frequency revealed that incorrect orientation values were not distributed randomly but were preferentially directed toward the ventricle, remaining parallel with the radial fiber scaffold (Supplementary Fig. 2b). These results indicate that a proper level of Wnt/β-catenin signal transduction is required for maintaining leading process stability and consequently the directional persistence during radial glia-guided locomotion.

Since the attachment of migrating pyramidal cells to radial glia is critical for leading process stability[21], we reasoned that altered adhesion may explain the above phenotype. We found that, while control cells adhered clearly to their adjacent nestin positive radial fiber, dnTCF4 overexpressing cells were presenting gaps between their membrane and the radial glia (Fig. 2d). To quantitatively assess attachment to glial fibers, we carried out a point-to-point measurement of the distance between migrating progenitor cells and nestin-positive radial fibers. The analysis revealed that the average distance between dnTCF4 overexpressing cells and fibers significantly increased compared to controls, in particular, in the region adjacent to the cell soma (Fig. 2d). It appears therefore that Wnt canonical signaling may contribute to leading process stability during locomotion by regulating the adhesion of migrating cells to radial glia.

We next explored potential downstream mechanisms that could mediate the effects of Wnt LOF on neuron-glia attachment. We characterized gene expression by bulk RNA sequencing on electroporated migrating neurons FACS purified at P0. (Supplementary Table 1). While common layer 2/3-specific markers expressed similarly in dnTCF4 samples and in GFP electroporated controls (Supplementary Fig. 3a) confirming the unchanged cellular identity, we identified 207 differentially expressed genes (DEGs; Fig. 3a), including well-known Wnt downstream targets (Supplementary Fig. 3b). Moreover, DEGs clearly separated dnTCF4 samples from controls on principal component analysis (PCA) plots (Fig. 3b). As expected, all migration-related gene ontologies from a publicly available database were significantly enriched in our gene selection (Supplementary Fig. 3c), which recognized 65 migration-related DEGs. The majority of these genes were strongly downregulated in the dnTCF4 cells (Supplementary Fig. 3d). One of them, C-Kit, a receptor protein-tyrosine kinase was of particular interest. Consistent with the marked downregulation of C-Kit expression after dnTCF4 overexpression (Supplementary Fig. 3e and see

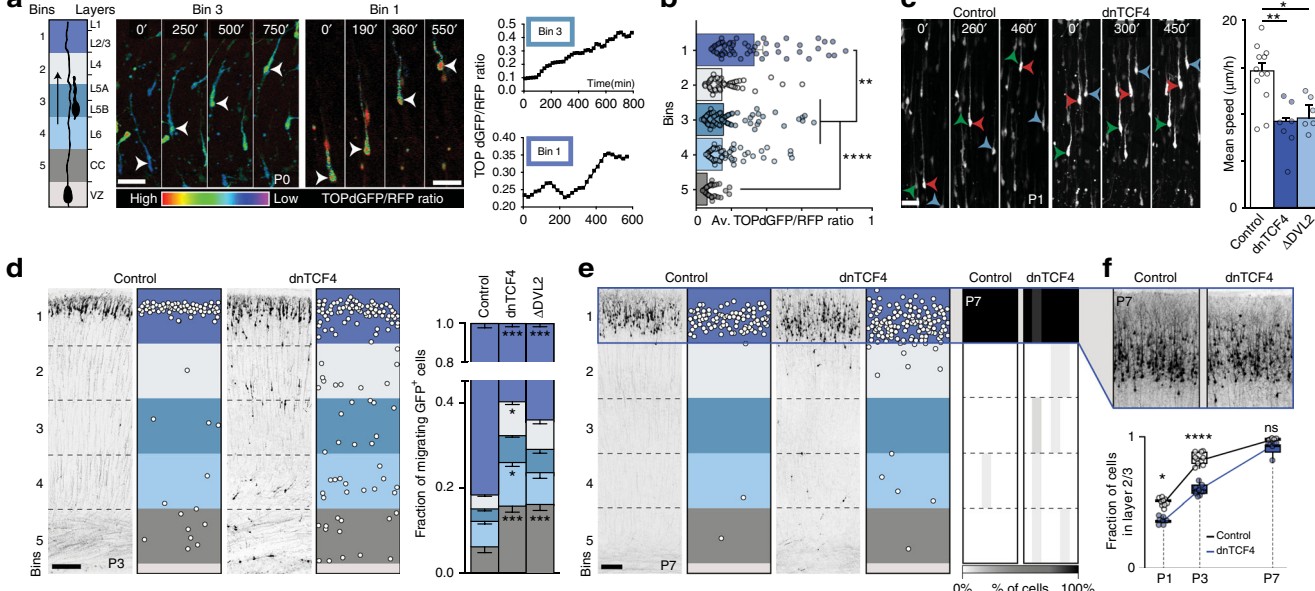

**Fig. 1** In vivo downregulation of canonical Wnt/ß-catenin signaling reduces migration speed and alters radial distribution of layer 2/3 CPNs in the rat somatosensory cortex. **a** Single-cell confocal imaging of TOPdGFP/UBI-RFP electroporated cells in the cortical plate. The migration path was divided in five equal bins corresponding to layers as illustrated. TOPdGFP/RFP intensity ratio shows dynamic Wnt-canonical signaling activity in neurons during radial migration at P0. The graphs represent the intensity ratio of the two illustrated cells over time. See also Supplementary Movie 1. **b** Averaged cellular ratios of TOPdGFP/RFP reveal increased Wnt-activity as cells approach the pial surface; n = 60, 47, 90, 65, and 38 cells, in bin 1, bin 2, bin 3, bin 4, and bin 5, respectively, Kruskal–Wallis. **c** Video time lapse images demonstrating that downregulation of canonical Wnt signaling (dnTCF4 and ΔDVL2) decreases speed of locomotion; quantification based on n = 13, 7, and 6 brains (Control, dnTCF4 and ΔDVL2, respectively), Mann–Whitney. See also Supplementary Movie 2. **d** Delayed dnTCF4-electroporated cells in P0 coronal sections. Fraction of migrating cells is shown in each bins; n = 10 brains per condition, two-way ANOVA. **e** By P7 delayed cells arrive to their layer. Heatmap of P7 cortical cell distribution at five levels, color-coded columns represent n = 6 brains/condition. **f** Comparable constitution of layer 2/3 in control and dnTCF4 brains at P7. The graph represent the proportion of cells arrived in the layer 2/3 at P1, P3, and P7 based on n = 9 and 7 brains at P1, n = 10 brains at P3 and n = 6 brains at P7 (Control, dnTCF4, respectively), Mann–Whitney. Graphs display mean ± s.e.m. *P < 0.05, **P < 0.01, ***P < 0.001, ****P < 0.0001. Bars = 20 μm (**a**, **c**), 50 μm (**f**) and 100 μm (**d**, **e**; overview)

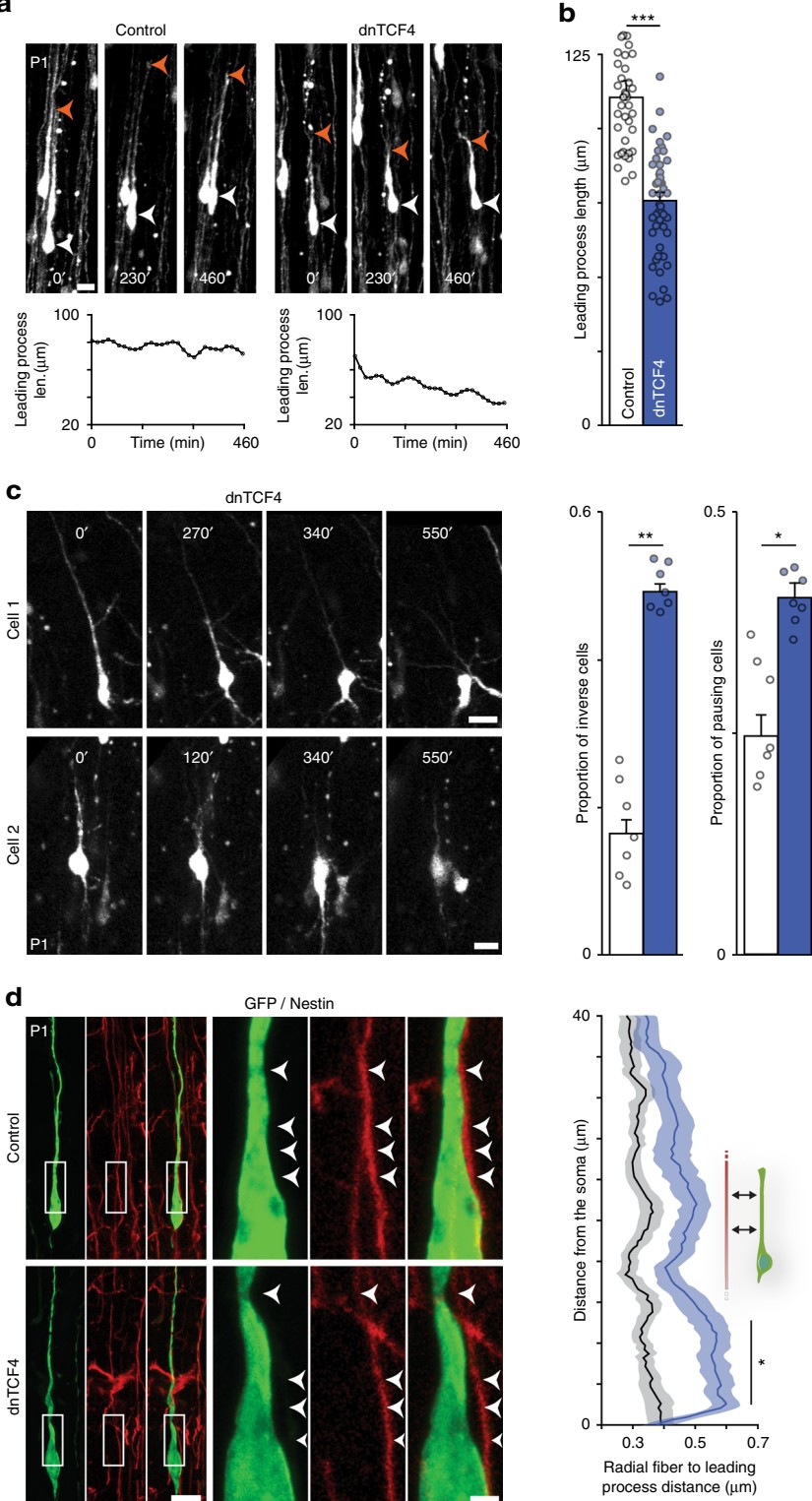

**Fig. 2** Loss of Wnt canonical signaling alters leading process stability and CPN attachment to radial glia. **a** Confocal time lapse sequence of representative control and dnTCF4 electroporated cells migrating in the cortical plate (CP). Graphs show dynamics of leading process length over time measured in the cells depicted. **b** Post hoc analysis of confocal images indicates that the leading process of dnTCF4 cells appears shorter than that of controls; $n = 50$ cells/condition from 7 brains, Student's $t$ test. See also Supplementary Movie 3. **c** Confocal time lapse sequences of two representative dnTCF4 cells migrating in the CP showing the frequently observable event of leading process inversion. Graphs represent the percentage of tracked cells exhibiting leading process inversion at least once (inverse cells) and the percentage of tracked cells remaining stationary for at least 100 min (pausing cells); $n = 7$ brains, Mann–Whitney. See also Supplementary Movie 4. **d** Representative images illustrating migrating cells attached to nestin-positive radial glia fibers. Note the increased space between the leading process of dnTCF4 electroporated cell and the glia fiber (arrowheads). We performed point-by-point measurement of the distance between the leading process of migrating cells and the radial fiber; $n = 50$ cells/condition from 4 brains, two-way ANOVA. Graphs display mean ± s.e.m. *$P < 0.05$, **$P < 0.01$, ***$P < 0.001$. Bars = 2 μm (**d**, close up) and 20 μm (**a**, **c**, **d**)

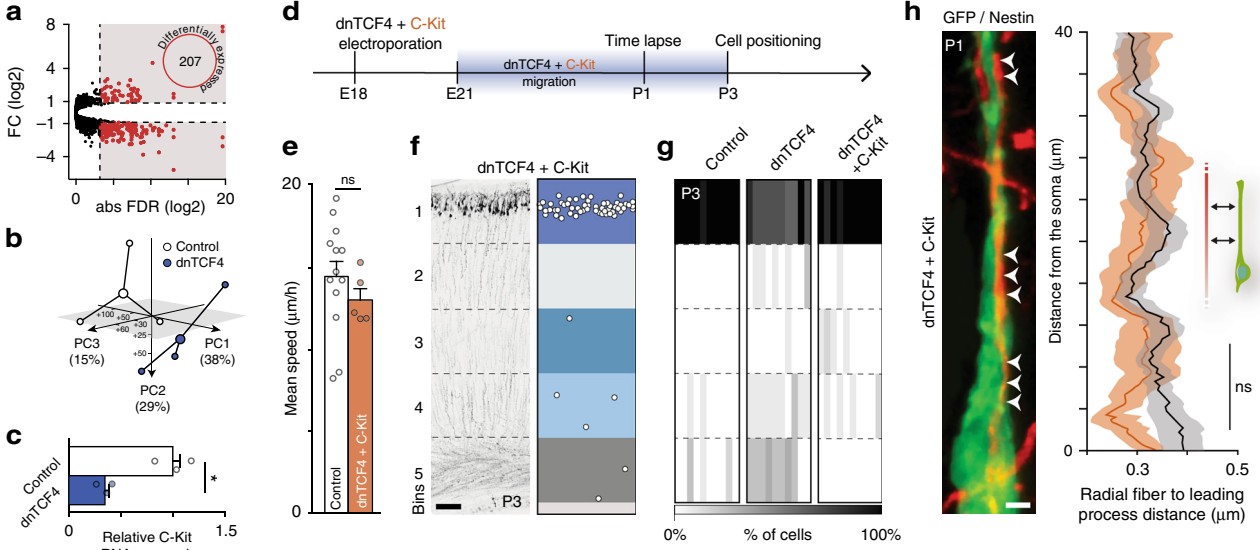

**Fig. 3** Wnt LOF perturbs leading process-radial fiber interactions via C-kit. **a** RNA sequencing identified 207 differentially expressed genes (DEGs) between control and dnTCF electroporated migrating neurons (selection by fold change (FC) of > 0.9 and < −0.9 and false discovery rate (FDR) of ≤ 0.1). **b** Three independent experiments plotted by principal components (PC) from PC analysis based on DEGs in control and dnTCF4 migrating neurons. See also Supplementary Table 1. **c** Relative C-Kit mRNA expression quantified by qRT-PCR on sorted migrating cells at P0 from control and dnTCF4 electroporated brains; $n = 3$ experiments, Student's $t$ test. **d** Timeline of rescue experiments with transient dnTCF4 and C-Kit co-expression during migration. **e** C-Kit overexpression during migration restores locomotion speed; $n = 13$ and five brains (Control and dnTCF4 + C-Kit, respectively), Mann–Whitney, $P = 0.1734$. **f** C-Kit overexpression reduces the proportion of delayed neurons represented in coronal section from a dnTCF4 + C-Kit electroporated brain at P3. **g** Heatmap of P3 cortical cell distribution, color-coded columns represent $n = 10$ brains per condition. **h** Representative image illustrating migrating cells attached to nestin-positive radial glia fibers. C-Kit overexpression in migrating cells restores a normal adhesion to the radial glia (arrowheads). Point-by-point measurement of the distance between the leading process of migrating cells and the radial fiber; $n = 50$ cells per condition from 4 brains, two-way ANOVA, $P = 0.1231$. Graphs display mean ± s.e.m. ns, non-significant, *$P < 0.05$. Bar = 2 μm (**h**) and 100 μm (**f**)

validation by qRT-PCR, Fig. 3c), we identified several TCF4 consensus sequences in its promoter (Supplementary Fig. 3f). Most importantly, C-Kit has recently been shown to have a robust effect on the migration speed of L2/3 neurons[22]. Hence, we tested the hypothesis that a decreased C-Kit expression could be responsible for the Wnt signaling related migration defect. We observed that over-expression of C-Kit specifically during the transient dnTCF4 overexpression (Fig. 3d) rescued the migration speed deficit (Fig. 3e) as well as the cell attachment to radial glia and restituted normal cell distribution in layer 2/3 at P3 (Fig. 3f–h and Supplementary Fig. 3g). Consistent with these results, we found that C-Kit overexpression alone significantly increased the migration speed and accelerated the positioning of neurons (Supplementary Fig. 3h, i). Together these data give strong support to the hypothesis that C-Kit is a downstream effector of the Wnt canonical pathway in regulating radial migration of pyramidal neurons.

**Late arriving CPNs develop altered callosal projections.** Next, we tested the hypothesis that delayed migration of CPNs might be associated with altered circuit development in the cerebral cortex. To investigate cortical circuits, we performed multi-electrode intracortical recording of somatosensory evoked potentials (SEPs) after unilateral whisker stimulation. Recording contralateral to whisker stimulation did not show differences in SEPs between control hemispheres and those exposed to transient dnTCF4 overexpression (Supplementary Fig. 4a, b). However, current source density analysis of SEPs recorded ipsilateral to whisker stimulation revealed a weaker and delayed response in the intact somatosensory barrel field if the contralateral homotypic cortex had been exposed to dnTCF4 during migration compared to

those exposed to control electroporation (Fig. 4a, b). Since this abnormal SEP response requires interhemispheric projections originating from neurons with altered migration, we reasoned that a dysfunctional inter-hemispheric communication between somatosensory cortices could underlie the SEP deficit. To test this idea indirectly, first we evaluated neuronal activity detecting the activity-reporter SAREdGFP[23] in layer 2/3 neurons of rats exposed to enriched environment after trimming all but two adjacent principal whiskers (C2, C3) contralateral to dnTCF4 or control electroporations (Supplementary Fig. 4c). Consistent with the electrophysiological data, we observed a reduced activity in the hemisphere contralateral to the electroporation following transient overexpression of dnTCF4 during migration (Supplementary Fig. 4c). Next, we compared callosal axons of normal and delayed neurons after the development of somatosensory callosal projections[24]. Axons of delayed neurons, while being normal ipsilaterally (Fig. 4c and Supplementary Fig. 4d), showed significantly reduced arborization in contralateral homotypic cortices, which is typically observed at the border of primary and secondary somatosensory areas (Fig. 4c, d). We also noted that the densities of axons in the ipsilateral cortex (Supplementary Fig. 4e) as well as those crossing the midline in the corpus callosum (CC) (Fig. 4c, d) were normal. Furthermore, the initial descent and callosal growth of these axons appeared unaltered on time-lapse videos (Supplementary Fig. 4f and Supplementary Movie 5). Thus, the reduced callosal connections of delayed neurons were not due to abnormal axonogenesis or axonal guidance, but rather caused by decreased contralateral branching and arborization. Importantly, Wnt activity returned to control levels during axonal arborization (i.e., P7-P14; Supplementary Fig. 4a), and transient overexpression of dnTCF4 after migration (P3-P7) did not decrease axonal arbor development

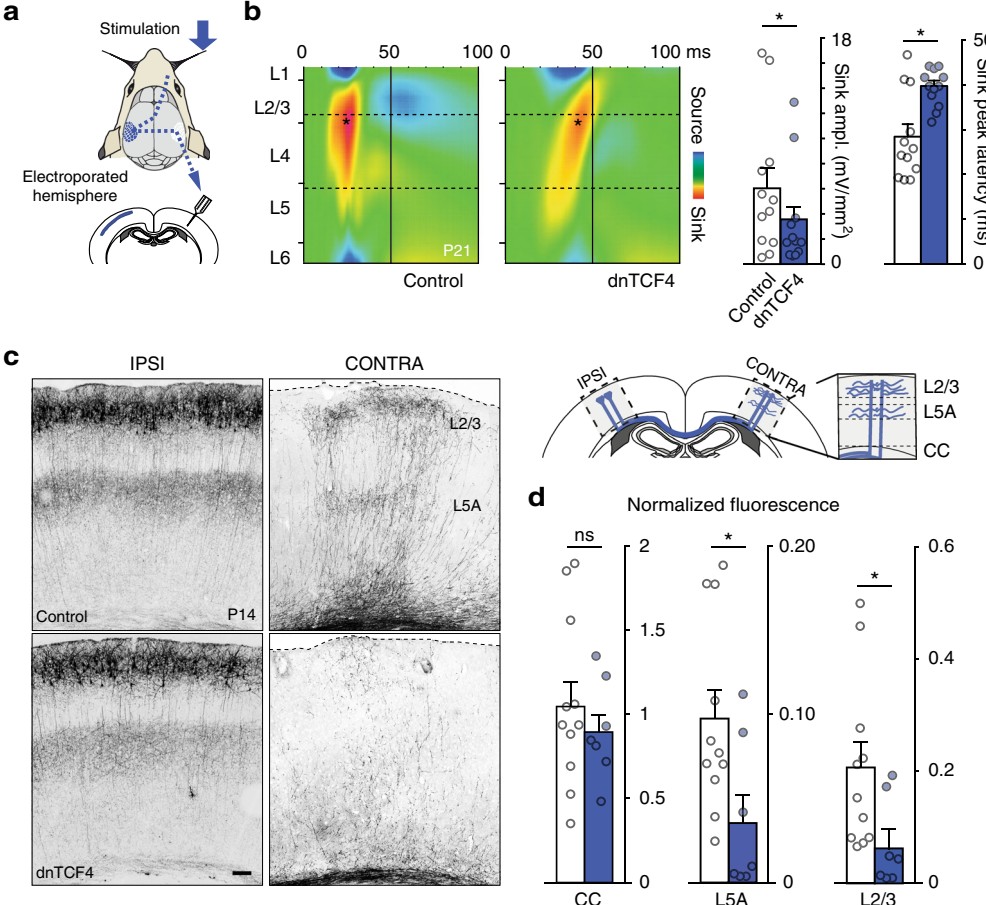

**Fig. 4** Delayed migration of CPNs is associated with abnormal interhemispheric connectivity. **a** Schematic of intracortical recording of whisker-stimulation evoked potentials. **b** Averaged current source density map of the somatosensory barrel field when recording ipsilateral to whisker stimulation. Graphs illustrate the quantifications of sink peak amplitude and peak latency; $n = 12$ animals, Wilcoxon test. **c** Schematic of CPN axons crossing in corpus callosum (CC) and arborizing in contralateral homotypic L2/3 and L5A. Late-arrived CPNs show preserved axonal morphology, with reduction of contralateral arbors at somatosensory cortices. **d** Average regional fluorescences normalized by ipsilateral L5A intensity; $n = 11$ and 7 brains (Control and dnTCF4, respectively), Mann–Whitney, $P = 0.4789$ for CC.

(Supplementary Fig. 5b). We conclude that the observed callosal axonal defect may not be a direct consequence of the reduced Wnt signaling, but could be related to the late arrival of CPNs.

**CPNs with migration delay exhibit reduced neuronal activity.** Strong evidence exists that neuronal activity plays a pivotal role in the formation of contralateral arbors of callosal axons[24, 25], therefore reduced activity could provoke the axonal abnormalities of late-arriving neurons. Using the SAREdGFP activity-reporter, we indeed observed a transient diminution of GFP labeling in the population of delayed cells compared to control neurons, present only at P10 but not at P21 (Fig. 5a). Electrical activity of neurons generally depends on cell-intrinsic electrophysiological properties and external afferent inputs. Using whole-cell patch-clamp recordings in acute slices, we excluded that transient dnTCF4 overexpression would lead to perturbed cell-intrinsic excitability of neurons during axonal arborization (Supplementary Fig. 6 and Table 1). Since early cortical networks emerge already among migrating neurons in radial columns[6], we next asked whether a reduction of afferent inputs from early cortical networks could account for the lack of neuronal activity at P10. Labeling excitatory connections by genetically coded PSD-95-specific fibro-nectin intrabodies (PSD-95.FingR-GFP)[26], we found a significantly decreased number of synaptic puncta in neurons with delayed migration (Fig. 5b). Consistent with this,

electrophysiological recordings of miniature excitatory post-synaptic currents (mEPSCs) of electroporated cells in acute slices from P14-P15 animals revealed that delayed neurons had significantly reduced mEPSC frequency compared to controls (Fig. 5c). To refine the origin of the missing connections, we used rabies-mediated monosynaptic retrograde tracing of layer 2/3 neurons during the above-mentioned developmental period (P7–P14). We found a general decrease in ipsilateral afferent inputs to late-arriving cells (Fig. 5d). This reduction was manifest in afferents from layer 4, a major peripheral entry towards layer 2/3 (Fig. 5e)[27]. Together, these findings are consistent with the hypothesis that a delay in migration is accompanied by an altered formation of cortical microcircuits and neuronal activation during early postnatal development, which consequently compromises the activity-dependent formation of transcallosal axons.

Finally, we wished to explore whether correcting reduced neuronal activity could reverse callosal structural defects of delayed neurons. We overexpressed a Gq-activator DREADD (hM3Dq) in late-arriving CPNs and temporally enhanced their activity during the critical period of axon arborization (i.e., P7-14; see experimental time line in Fig. 5f). Postnatal modulation of the activity of CPNs prevented the axonal defect (Fig. 5f). Thus, effects of perinatal perturbations of neuronal migration on interhemispheric circuits can be rescued by enhanced activation of these neurons during the critical postnatal period.

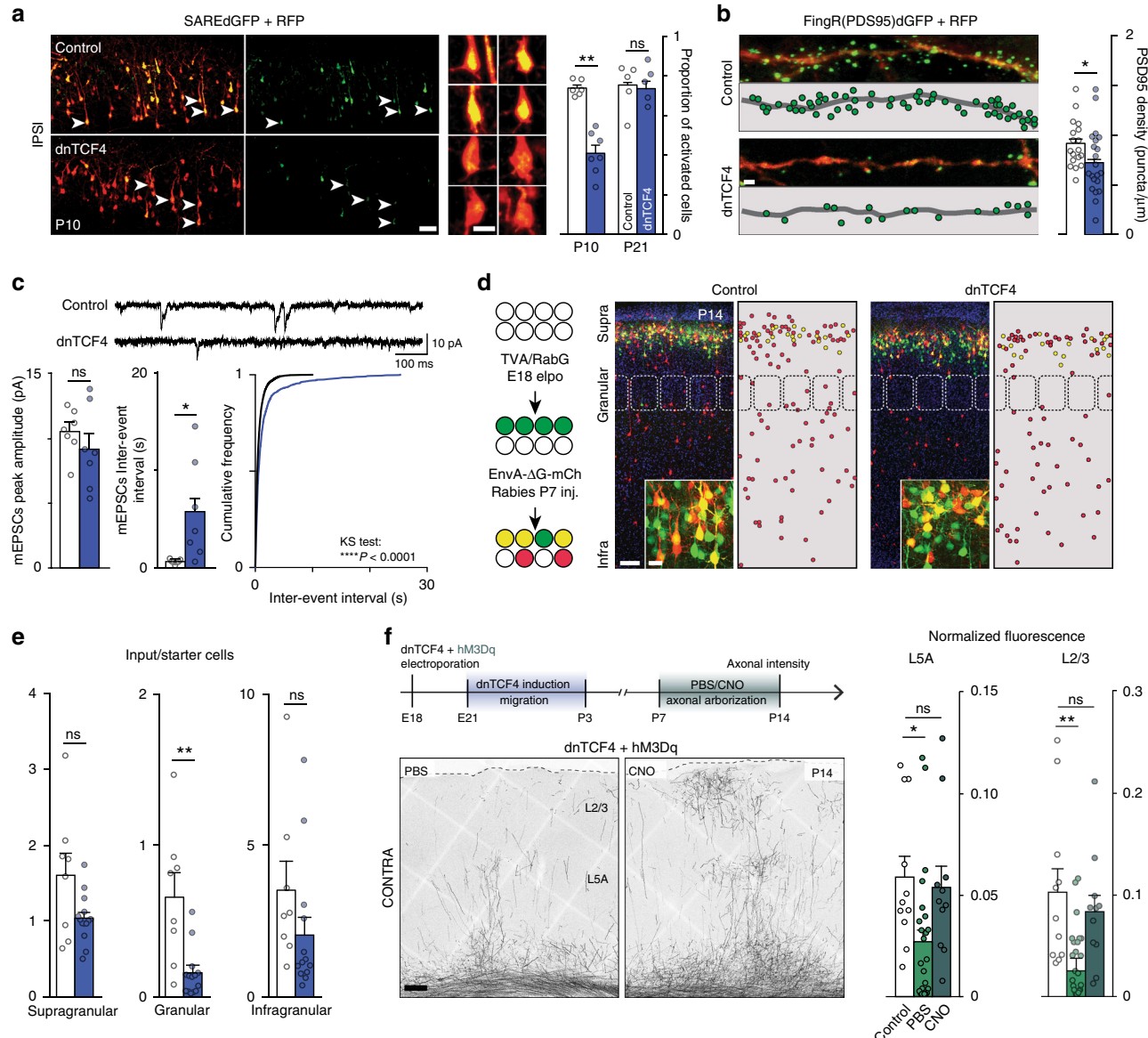

**Fig. 5** Delayed callosal projection neurons display reduced synaptic input and reduced neuronal activity. **a** Activity-reporter (SAREdGFP) shows decreased GFP expression in late-arrived neurons (arrowheads mark activated cells, magnified at right). Graph shows the proportion of GFP-positive activated cells during axonal arborization (P10) and at P21; $n = 6$ and 7 brains (Control vs. dnTCF4, respectively) at P10 and $n = 5$ brains at P21, Mann–Whitney, $P > 0.9999$ for P21. **b** Excitatory connections labeled by PSD-95-specific FingR-GFP showing a decreased number of synaptic puncta in neurons with delayed migration (dnTCF4); $n = 18$ and 21 cells from 3 and 4 brains (Control vs. dnTCF4, respectively), Mann–Whitney. **c** mEPSCs recording samples of layer 2/3 pyramidal neurons for control (top trace) and dnTCF4 (bottom trace) P14-P15 animals. No significant difference in amplitude and a significant increase in mEPSCs inter-event interval for dnTCF4 animals; $n = 7$ animals per group, 1 cell per animal, Student's $t$ test (bar graphs, $P > 0.9999$ for P21), Kolmogorov–Smirnov test (cumulative distribution). **d** Rabies-mediated monosynaptic retrograde tracing (schematized at left) shows reduced afferent inputs on late arrived (dnTCF4) starter cells, magnified inserts display equal density of starter cells. **e** Quantification of input/starter cells shows a major reduction of inputs from the granular layer; $n = 8$ and 14 brains (Control and dnTCF4, respectively), Mann–Whitney, $P = 0.1876$ for supragranular layer. **f** The timeline showing the experimental design used for rescuing aberrant callosal projections via hM3Dq DREADD receptor activation. Daily clozapine-n-oxide (CNO) injections were performed during axonal arborization. Chemogenetic stimulation of late-arrived CPNs rescues contralateral arbors compared to non-activated neurons (PBS injection); $n = 11$, 24, and 11 brains (Control, PBS, CNO, respectively), Kruskal–Wallis, Control vs. CNO: $P = 0.5455$ for L5A $P > 0.9999$ for L2/3

**Long-term alteration of sensory function and social behavior**. Since complete migratory arrest of a small fraction of neurons in the dorsomedial cortex have been shown to be sufficient to perturb animal behavior[28], we set out to test whether the transitory delay in the dorsolateral somatosensory cortex could also generate long-term sensory deficits or related behavioral alterations in adulthood. First, we tested haptic skills and fine sensorimotor coordination in young adult rats after somatosensory CPN migration delay (referred to as dnTCF4 animals; see experimental time line in Fig. 6a). Adult dnTCF4 rats (1) needed more whisking trials and hesitated longer before crossing the experimental gap (Fig. 6b, left), (2) made significantly more step errors (i.e., slipped into variably-sized grid-holes) when running on irregular grid (Fig. 6b, middle), and (3) were slower in

**Table 1 Intrinsic electrophysiological properties of layer 2/3 neurons at P14**

| | Control (n = 8 cells) | | dnTCF4 (n = 11 cells) | | P-value |
| --- | --- | --- | --- | --- | --- |
| | Mean | s.d. | Mean | s.d. | |
| Resting membrane potential (mV) | −58.68 | 11.58 | −62.2 | 7.99 | 0.4423 |
| Input resistance (MOhm) | 35.1 | 14.96 | 26.04 | 9.491 | 0.1241 |
| Tau (ms) | 33.38 | 14.58 | 32.45 | 8.134 | 0.8596 |
| Sag (mV) | 1.235 | 1.537 | 0.4708 | 0.4513 | 0.8855 |
| Spike threshold (mV) | −30.06 | 4.895 | −29.55 | 5.781 | 0.8402 |
| Rheobase (pA) | 83.23 | 72.14 | 99.85 | 58.19 | 0.5853 |
| Spike peak amplitude (mV) | 58.45 | 11.66 | 56.22 | 8.986 | 0.643 |
| Spike half width (ms) | 2.326 | 1.029 | 2.197 | 0.412 | 0.7452 |
| First interspike interval (ms) | 19.63 | 14.12 | 14.96 | 4.243 | 0.428 |
| Last interspike interval (ms) | 47.71 | 20.65 | 44.44 | 16.84 | 0.7403 |
| Adaptation (%) | 50.43 | 13.01 | 55.25 | 8.396 | 0.4029 |

To exclude the possibility that transient dnTCF4 overexpression may impair intrinsic electrophysiological properties of delayed CPNs, we compared 11 electrophysiological parameters in dnTCF4-electroporated neurons with control electroporated cells and did not find significant differences among the groups; n = 8 and 11 cells (Control and dnTCF4, respectively), Student's t test. Thus, reduced activity of delayed CPNs is not related to detectable modification of intrinsic electrophysiological properties

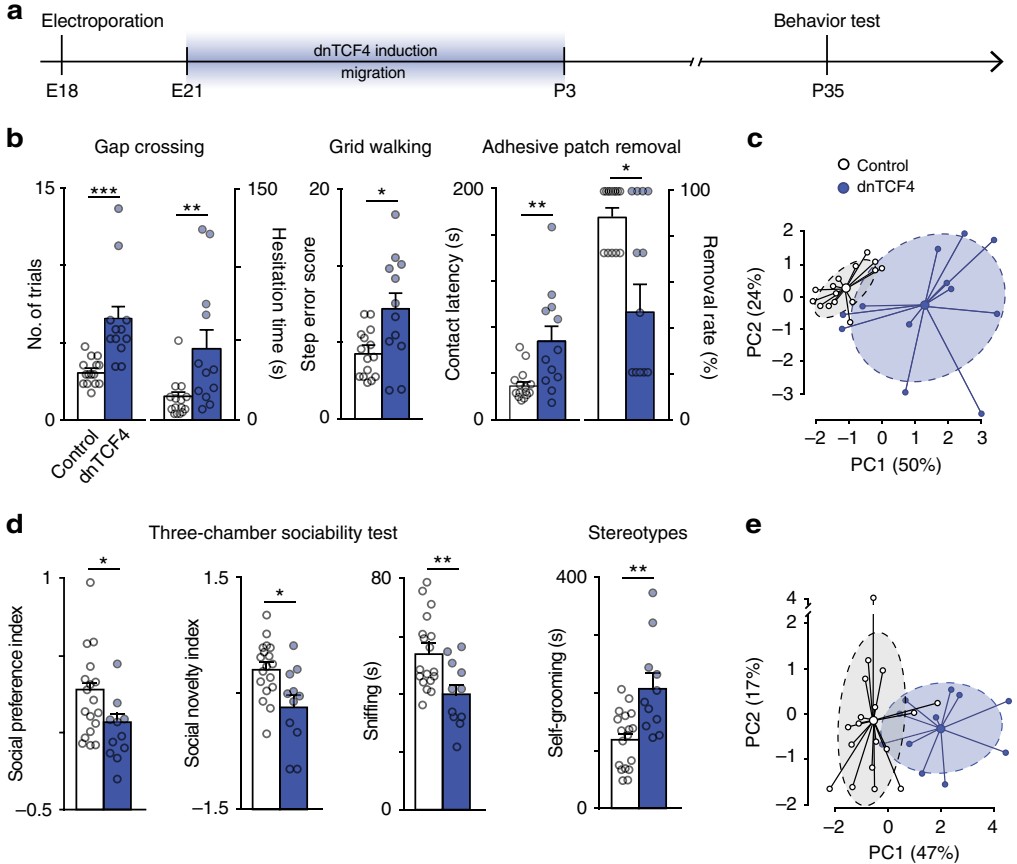

**Fig. 6** Late arrival of CPNs leads to long-term deficit of sensory function and social behavior. **a** Experimental timeline. **b** Sensory-oriented behavior tests show deficit in adult rats with delayed CPN migration; n = 15 and 12 animals (Control and dnTCF4, respectively), Mann–Whitney. **c** Principal component (PC) analysis of variables from sensory tests separates animals with late-arrived neurons (dnTCF4) from controls. **d** Decreased sociability and interaction (sniffing) as well as increased tendency for repetitive movements (self-grooming) in animals with late-arrived CPNs; n = 18 and 11 animals (Control and dnTCF4, respectively), Mann–Whitney. **e** Plots of PC analysis of autistic-like variables. Graphs display mean ± s.e.m. *P < 0.05, **P < 0.01, ***P < 0.001

recognizing a patch on a hind paw, as expressed by contact latency (Fig. 6b, right), in the particularly sensitive patch-removal test[29]. Together these results indicate impaired somatosensation. This conclusion received strong support from the analysis of the area distribution of electroporated cells in animals used for behavioral tests. We found that our electroporation mainly affected the primary somatosensory cortex and primary and

secondary motor areas, M1 and M2, were much less involved (Supplementary Fig. 7a). Importantly, the lack of detectable deficit in spontaneous locomotor activity during open-field tests (Supplementary Fig. 7b) did not support associated locomotor deficiency either. As dnTCF4 animals clustered apart from controls by PCA on behavioral variables (Fig. 6c), our data were consistent with a general deficit in somatosensory function.

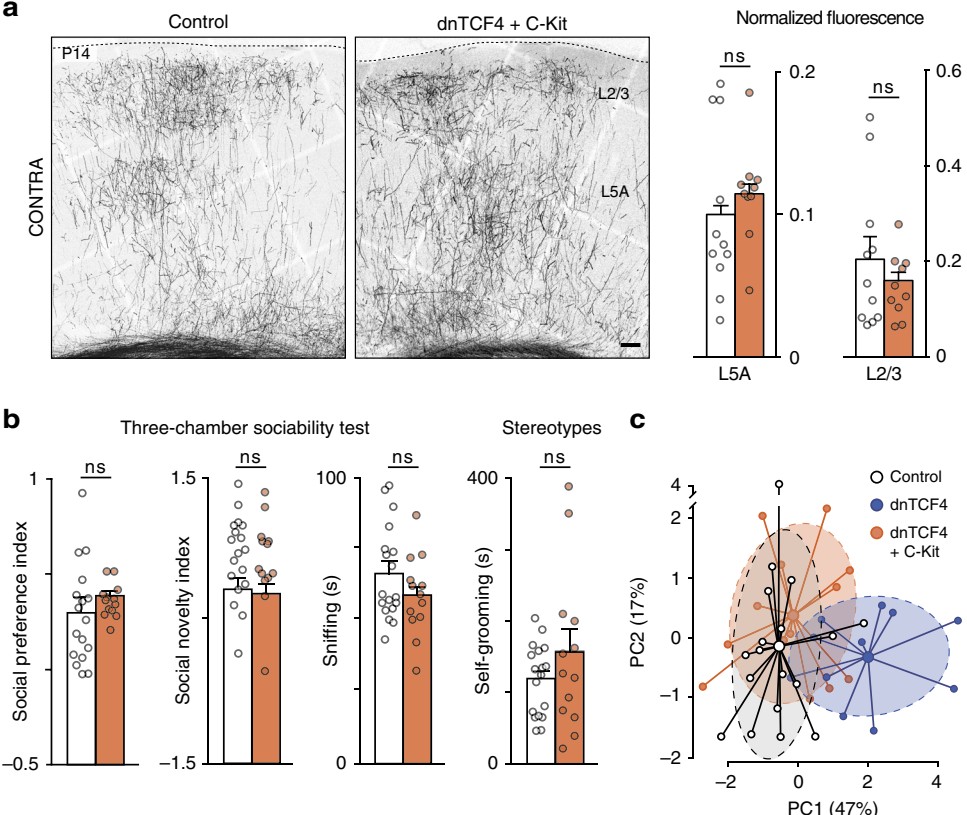

**Fig. 7** Correcting CPN migration delay rescues the circuit and behavior phenotype. **a** Restored migration by C-Kit overexpression rescues axon arborization. Graphs show the average regional intensity in L5A and L2/3, $n = 11$ and 10 brains (Control and dnTCF4 + C-Kit, respectively), Mann–Whitney, $P = 0.1971$ for L5A and $P = 0.7564$ for L2/3. **b** Sociability, interaction (sniffing), and tendency for repetitive movements (self-grooming) are rescued by C-Kit overexpression; $n = 18$ and 13 animals (Control and dnTCF4 + C-Kit, respectively), Mann–Whitney, $P = 0.1324$ for social preference, $P = 0.7004$ for social novelty, $P = 0.3167$ for sniffing and $P = 0.5736$ for self-grooming. **c** PC analysis plots of autistic-like variables.

Since somatosensory perturbations are part of the core ASD symptoms[30, 31] and recently were shown to be directly responsible for altered development of social functions in ASD mouse models[32], we postulated that dnTCF4 animals might also display autistic-like social and compulsive defects. To test this hypothesis, we performed the three-chamber sociability test and evaluated repetitive movements. In comparison with control animals, rats with the migration deficit showed significantly less social interest, and more repetitive movements (i.e., self-grooming, Fig. 6d), recapitulating characteristic behavioral components of ASD. Furthermore, PCA plots displayed only small overlap between control and dnTCF4 groups, confirming a general change in ASD-like behavioral aspects, as well (Fig. 6e). Taken together, these results demonstrate that a transient downregulation of canonical Wnt signaling and the subsequent perturbation of the migration and connectivity of a subpopulation of CPNs is sufficient to produce long-term somatosensory deficit and altered social behavior.

**Rescue of connectivity and behavior phenotypes**. To corroborate the interpretation that the precise timing of migration is critical for specifying circuit formation and behavior, we examined whether rescuing migration errors via simultaneous C-Kit overexpression could also rescue cortical circuits defects and prevent the development of autistic-like behavior. Restitution of normal migration with C-Kit overexpression normalized transcallosal axon development as well (Fig. 7a). Furthermore, dnTCF4 + C-Kit animals exhibited improved social behavior and

reduced self-grooming (Fig. 7b), and on behavioral PCA plots, they clustered largely, overlapping with control animals (Fig. 7c). Taken together, these results support the idea that perturbed neuronal migration may lead to altered callosal connectivity and behavior deficits.

Given that upregulation of neuronal activity during the critical developmental period of callosal connectivity can rescue the axonal phenotype, we reasoned that activity could also rescue the behavioral phenotype. To test this idea, we have co-electroporated hM3Dq with dnTCF4 and artificially enhanced the activity of late-arriving CPNs during the period of axonal arborization (i.e., P7-14). We have found that >2 weeks after activity enhancement all variables concerning social and compulsive behavior were normalized in such animals (Supplementary Fig. 7c). This was corroborated also by the clear overlap between control and hM3Dq groups on behavioral PCA plots (Supplementary Fig. 7d). These results support our hypothesis that a deficit in activity-dependent formation of callosal axon arbors plays a key role in impaired social and compulsive behavior.

## Discussion

In the present study, we demonstrate how transient disruption of canonical Wnt signaling leads to alterations of neuronal migration, disturbed circuit development and impaired behavior. While it has been speculated about such a relationship, direct evidence was so far lacking. The discovery of a new role for canonical Wnt signaling in regulating the pace of glia-guided radial migration permitted to establish a new animal model of migratory delay

without persisting migration arrest. We find that delayed neurons receive less afferent input, exhibit reduced neuronal activity during critical postnatal periods and develop aberrant callosal projections. We demonstrate that just a subset of CPNs with altered migration and connectivity leads to long-term behavioral abnormalities ranging from sensory dysfunction to social deficits and compulsive behavior. These phenotypes can all be rescued by normalizing migration with a Wnt canonical downstream element, C-Kit. Enhancing the activity of neurons during a critical postnatal period restores normal morphology of callosal axons and rescues the behavioral phenotype of delayed migration. These findings might start to answer long-standing questions regarding the role of migration errors and altered cortical circuit development in the pathogenesis of neuropsychiatric diseases.

Previous studies have recognized the importance of the evolutionary conserved canonical Wnt/β-catenin signaling in early corticogenesis, controlling proliferation as well as differentiation of neuronal progenitors[13, 14]. β-catenin gain-of-function leads to massive expansion of the progenitor pool[33] and LOF causes premature exit from the cell cycle, migration and neuronal differentiation[34]. Interestingly, upregulation of Wnt signaling by Wnt3A overexpression drives early differentiation of intermediate progenitors into neurons, thereby provoking neuronal heterotopias[15]. Here we show that adequate levels of canonical Wnt signaling activity are also essential for the proper locomotion of neurons during radial migration. Our single-cell imaging revealed that canonical Wnt activity is high during radial glia-guided migration in the upper cortical plate. Moreover, we found that decreasing Wnt signaling in CPNs resulted in reduced migratory speed and delayed arrival. Migrating neurons with reduced Wnt signaling adhered less to radial glia fibers and exhibited shortened and frequently inverted leading processes. Thus, Wnt/β-catenin signaling pathway contributes to the regulation of leading process stability, which is critical for maintaining the adequate tempo and the directional persistence of radial glia-guided migration. This represents a new function of canonical Wnt signaling in cortical development, extending the range of its previously identified roles including proliferation[35–37] differentiation[15, 38, 39], switch from progenitor proliferation to migration[17] and multipolar-to-bipolar transition of cortical progenitors[18].

We identify C-Kit as a potential downstream effector of the Wnt canonical pathway in regulating locomotion during radial migration. The tyrosine kinase receptor C-Kit and its ligand, stem cell factor (SCF), are well-known mediators in proliferation, survival, and positive chemotaxis of different cell types[40]. Our results are consistent with the previous data showing SCF being expressed in an inside-out gradient pattern during the development of the cerebral cortex (i.e., low in deep layers 4–6 and high in superficial layers 2–3)[22]. Knockdown of C-Kit in migrating L2/3 pyramidal neurons perturbs their migration, leading to a delayed radial migration[22]. On the other hand, C-Kit overexpression leads to faster migration of cortical L2/3 neurons and in vitro application of the kit ligand SCF also enhances the migration of neurons[22]. Our study not only lends strong support to these findings, but also extends them by establishing that canonical Wnt signaling acts upstream of the C-Kit pathway. Thus, Wnt/C-Kit signaling, similar to connexins[21], integrins[41, 42] and N-cadherin[43] contributes to the regulation of neuron-glia adhesive interactions, which is a key determinant of glia-guided neuronal migration. Previous studies have demonstrated that C-Kit has a dual role; its interaction with the SCF ligand activates intracellular signaling, and serves as mechanical anchorage as well[44]. It remains to be determined whether one of these functions or both are involved in radial migration.

Interfering with Wnt signaling during embryonic development of the cerebral cortex can cause multiple defects in diverse arrays of developmental steps[13]. Here we demonstrate that a transient decrease of canonical Wnt signaling during glia-guided locomotion delays but does not permanently arrest migration of CPNs, and these neurons with Wnt LOF eventually reach their final destination in layer 2/3. It has been proposed that abnormal neuronal migration contributes to the pathogenesis of neurodevelopmental disorders such as ASD, which is characterized by poor social interactions, repetitive behavior and altered communication[8, 11, 12, 45]. In support of this hypothesis, we show that delayed migration of a subset of CPNs is associated with long-term behavioral abnormalities ranging from sensory dysfunction to ASD-like social deficits and compulsive behavior. Our results are consistent with the observations that defects in sensory information processing are frequently associated with behavioral alterations such as stereotyped movements and social impairment[30–32, 46]. Cortical heterotopias observed in humans with ASD indicate a complete arrest of neuronal migration during development[8, 11, 12, 45]. Our results support that even a transitory delay of only a subset of CPNs could be sufficient to cause similar network and behavioral alterations. However, the relevance of these findings to human disease remains to be demonstrated.

How does transient downregulation of Wnt canonical signaling lead to abnormal behavior? We found that neurons with delayed positioning develop aberrant axonal projections that run in the CC, the major fiber bundle connecting the two cerebral hemispheres[10, 47, 48]. Importantly, the observed changes are specific for the axonal projections since these neurons exhibit normal dendritic arbor development. Delayed neurons also display reduced neuronal activity and postnatal chemogenetic enhancement of neuronal activity is sufficient to rescue interhemispheric projections. These results are consistent with earlier studies showing that the development of callosal connections is an activity-dependent process[24] and demonstrate for the first time that altered migration may impact on it. We speculate that late arrival generates competitive disadvantage to recruit afferent inputs. Supporting this hypothesis, we identified a reduction in short-range afferent connections, suggesting the perturbation of synaptic integration and subsequent activation of delayed neurons. Together, these findings lend support to the concept that neuronal migration and microcircuit assembly in the cerebral cortex are closely linked[6]. More specifically, our study suggests a link between migration delay, deficient callosal connectivity, and the genesis of social impairment as well as compulsive behavior by demonstrating that restoring delayed migration by transient overexpression of the Wnt-downstream effector C-Kit prevented circuit and behavior alterations. It should be noted, however, that we cannot exclude the possibility that long-term changes, unrelated to the migration of neurons and undetected in the present study after a transient decrease of Wnt/C-Kit signaling could also contribute to the observed alteration in synaptic input formation, neuronal activity and callosal axon arborization. Additional studies, involving alternative means to delay neuronal migration, independent of Wnt signaling, will be needed to give definitive answer to this question.

Emerging evidence suggests that in humans aberrant connectivity in the neocortex, as well as in subcortical structures, underlies the clinical manifestations in ASD[48–50]. For example, agenesis of the CC has been recognized as a major risk factor for ASD[47] and recent functional magnetic resonance and diffusion tensor imaging studies have revealed significant reductions in size of the CC in a subgroup of ASD patients[51, 52]. Besides, defects in long range callosal connections have been observed in the Fmr1 mouse model of autism as well[53]. A more precise analysis of transcallosal connections across multiple ASD patient data sets revealed a reduced homotypic interhemispheric connectivity of primary sensory cortical areas in most autistic patients[48]. Our

findings provide insights into a disrupted Wnt signaling-related migratory error that could be associated with analogous alterations in interhemispheric connectivity. Whether such a mechanism could also be involved in the pathogenesis of human diseases remains to be determined. It is of interest, however, that Wnt-pathway dysregulation has also been associated in humans with ASD[54, 55]. How dysregulated Wnt signaling in these cases might lead to behavioral or neurological symptoms remains a matter of debate.

Developing mechanism-based therapies for neurodevelopmental disorders is a major challenge. Here we show that reduced arborization of callosal axons and impaired behavior after prenatal perturbations of canonical Wnt signaling and neuronal migration could be corrected by enhanced activation of these neurons during early postnatal periods. Considering that, in humans, the maturation of interhemispheric connections continues during childhood[56], our results might open new perspectives for research aimed at correcting structural alterations of interhemispheric connections via activity-based approaches[57].

## Methods

**Animals.** All animal procedures were conducted in accordance with the Swiss laws, and previously approved by the Geneva Cantonal Veterinary Authority. Wistar rats were provided by Charles River Laboratories. Animals of both sexes were used at the indicated ages. Due to the electroporation procedure, no randomization was possible for the selection of animals into control or experimental groups. Control and experimental animals were housed in mixed cages, in the institutional animal facility under standard 12 h/12 h light/dark cycles, with the maximum number of animals per cage surface housed together, determined by the institutional animal facility regulations. Where indicated, doxycycline (Sigma) was administered (1 mg/ml) in drinking water at embryonic day 21 (E21) in order to activate transgene expression from inducible constructs. For chemogenetic activity restoration experiments, intraperitoneal injections of Clozapine-N-oxide (CNO, 100 µl, 10 mg/ml) were performed daily from postnatal day 7 (P7) until P14 in order to activate the engineered G-protein coupled receptor, hM3Dq.

**Plasmids.** The different expression vectors were produced using the Gateway® recombination cloning technology[58]. A first class of constructs contains a single expression unit, with one promoter governing the transcription of one single gene, as in pCLX-UBI-GFP (GFP, #27245, Addgene), pCLX-UBI-Tred (RFP, #27246, Addgene) and pCLX-M38TOP-dGFP (TOPdGFP). Further details on these plasmids can be found in ref. [18]. TET-controlled auto-inducible expression vectors, derived from the Polyswitch Lentivector system described previously[58], were used for transient expression during neuronal migration. They contained the rtTA transactivator fused to either GFP or BSD expressed form a constitutively active promoter (PGK or EFs), and the Gene of Interest (GOI) under the control of the pTF optimized TET-responsive promoter. Details on these plasmids can be obtained at Addgene.com (plasmids #45952 and #45953). GOIs used in this auto-inducible vector were dnTCF4 (pCWX-pTF-dnTCF4-PGK-GFP-E2A-rtTA), C-Kit (a kind gift from Dr. B. Wehrle-Haller, pCWX-pTF-cKit-PGK-GFP-E2A-rtTA), ΔDVL2 (pCWX-pTF-ΔDVL2-PGK-GFP-E2A-rtTA), shWnt3A (pCWX-pTF-shWnt3A-PGK-GFP-E2A-rtTA), Wnt1 (pCWX-pTF-Wnt1-PGK-GFP-E2A-rtTA), Wnt3A (pCWX-pTF-Wnt3A-PGK-GFP-E2A-rtTA) or Δ45β-catenin (pCWX-pTF-Δ45β-catenin-PGK-GFP-E2A-rtTA). dnTCF4 is a dominant-negative form derived from TCF4 transcription factor, which has a deletion of the NH2-terminal 30 amino acids important for the interaction with β-catenin. The resulting deletion mutant binds DNA at a specific Wnt response element but lacks β-catenin-binding site, therefore it represses transcription of canonical Wnt signaling target genes[19]. Repression of Wnt signaling was also obtained by over-expression of a mutated form of Disheveled 2 (ΔDVL2)[20]. This construct code for the protein Disheveled-2, which lacks the DIX domain important for canonical Wnt signaling activation. For Wnt3A knockdown experiments, we used an optimized 1-loop miRNA-based design (shWnt3A)[59] targeting the following sequence: 5′-GCAGGAACTAAGTGGAGATCAC-3′. The activity reporter AVV-SARE-GFP construct (SAREdGFP) was the courtesy of Dr. D. Muller[60]. This plasmid encodes a bidirectional expression cassette that expresses a red fluorescent protein (TurboFP635) under the control of the constitutive promoter PGK and a destabilized version of the GFP (d2EGFP) under the control of a synaptic activity responsive enhancer (SARE) previously described in ref. [61]. TVA (pCMMP-TVA800, #15778, Addgene) was a kind gift from E. Callaway[62], the RabG (pHCMV-RabiesG, #15785, Addgene) expression plasmid was a kind gift from M. Sena-Esteves[63]. For the chemogenetic activity restoration experiments, we used a Mifepristone-controlled auto-inducible plasmid expressing the hM3Dq gene. This choice of inducibilty resulted from preliminary experiments, where we observed a certain level of basal activity of the hM3Dq gene even in the absence of its CNO ligand

(data not shown). Since our experiments already involved gene induction using DOX, we needed another gene switch independent of DOX. We thus adapted the Gene Switch available from Invitrogen (ThermoFisher Scientific, Ref. K106002), which responds to Mifepristone and constructed a Gateway-compatible plasmid containing all the components for auto-induction together with a Cherry reporter gene. The resulting plasmid pCLX-MIFE-hM3Dq_PGK-mCherry (hM3Dq) was recloned from the pCDNA5/FRT-HA-hM3D(Gq) plasmid, (#45547, Addgene, a kind gift from B. Roth). Further details on the plasmids and the cloning procedures that have been used for their preparation can be obtained at http://lentilab.unige.ch/.

**Luciferase reporter assays.** These assays were carried out according to standard protocols[64]. Briefly, HEK293T cells were plated at $5 \times 10^4$ cells per well 24 h before transfection. The following plasmids were used: Prl-TK (Promega) as transfection control, Δ45β-catenin, Wnt1 and Wnt3A (for different levels of Wnt canonical activation), TOPflash (TCF Reporter Plasmid with wild-type TCF-binding sites, Millipore), and FOPflash (mutant TCF binding site, Millipore) as negative control. Luciferase activity was measured following the Dual Luciferase Assay protocol (Promega) at 24 h post-transfection with a Victor3 Multilabel Reader 1420-015.

**In utero electroporation.** In utero electroporation was performed at E18 to target layer 2/3 pyramidal neurons, as previously described[18]. Briefly, pregnant rats were anesthetized using Isofluran (Foren 2%) in a mix of 30% of oxygen and 70% of air. About 1 µl of DNA solution (with a maximum plasmid concentration of 5 µg/µl) was injected through the uterine wall into the lateral ventricle of the fetal brains. The head of embryos was placed between forceps platinum electrodes ($d = 0.5$ cm, NepaGene, CUY611P3-1) in order to target the developing somatosensory cortex. An electric field was generated using the square waved electroporator (NepaGene, CUY21SC) with the following settings: 50 mV voltage, 50 ms exposure time, 5 pulses at 1 Hz. For some purposes, electroporation was performed bilaterally. In these cases after classical electroporation of one hemisphere with the first (control) plasmid, the second (experimental) plasmid was injected and electroporated in the contralateral hemisphere.

**Immunofluorescence and in situ hybridization.** Under deep pentobarbital anesthesia animals were perfused transcardially with 0,9% NaCl followed by 4% paraformaldehyde (PFA). Brains were post-fixed in 4% PFA for one or 2 days, depending on brain size. Then either 20 µm thick cryostat or 50 µm thick vibratome (for axonal intensity measurements and dendritic reconstruction) coronal sectioning was performed. For cryostat sectioning, the brains were cryoprotected in 20% sucrose for 24 h. Immunofluorescence was carried out as follows: after 1 h pre-incubation with PBS/0.5% bovine serum albumin (BSA)/0.3% Triton X-100 at room temperature (RT), slides where incubated overnight at 4 °C with the primary antibody diluted in the same solution. The following primary antibodies were used: goat anti-GFP (1:1000, Novus Biologicals, NB 100-1770), rabbit anti-GFP (1:1000, Millipore, AB3080), mouse anti-GFP (1:5000, Novus Biologicals, NB 600-597), mouse anti-Satb2 (1:200, Abcam, AB51502), rabbit anti-S100B (1:1000, Swant, 37 A). Subsequent incubation with the appropriate secondary antibody was done for 90 min at RT. The following secondary antibodies were used: anti-rabbit Alexa-488, Alexa-568, Alexa-647; anti-mouse Alexa-488, Alexa-568; and anti-goat Alexa-488. For nuclear counterstaining, 15 min at RT incubation with Hoechst (Invitrogen, Molecular probes) was performed. In situ hybridization was performed as previously described in ref. [18].

**Enriched environment.** For the activity measurement of contralateral homotypic layer 2/3, neurons an exposure to enriched environment was performed on bilateral electroporated animals: dnTCF4 was electroporated in the left hemisphere and the activity reporter construct (SAREdGFP) in the right. At P35, we cut all but two whiskers (C2 and C3) in the left snout. Enriched environment was performed in a 1406 cm² large, transparent plastic cage, with multiple objects placed at random, but fixed locations inside the cage. Rats were allowed to explore the cage for 6 h in the dark, and were sacrificed and perfused immediately after.

**Intracortical recordings.** We recorded intracortical local field potentials (LFPs) in P21 Wistar rats under light isoflurane anesthesia (initiation and surgery at 2.5%, recordings at 0.8–1.1%) in a mix of 30% of oxygen and 70% of air). Rats were mounted in a stereotaxic frame and a subcutaneous dose of bipuvacaine was injected above the skull before we incised the skin on the midline from the frontal to the occipital pole and retracted it laterally. Two large craniotomies were performed in the right and left parietal bones to expose the lateral parietal cortices. Linear 16-electrode probes (iridium based, 177 µm² electrode diameters, 100 µm inter-electrode spacing, NeuroNexus Technologies) were inserted perpendicular to the cortical surface into S1 barrel cortex regions of both hemispheres (1.5 mm caudal and 5 mm lateral to bregma). Before insertion, probes were painted with a fluorescent marker (Dye I, Invitrogen) to recover their positions via histology. After insertion, the surface of the brain was covered with sterile saline solution heated at 37 °C. Differential signals against a skin reference electrode were acquired with the multichannel Digilynx system (Neuralynx). The LFP signal was bandpass-filtered online between 1 and 4000 Hz, and sampled at 8 kHz. Unilateral whisker stimuli

were delivered during recordings through a solenoid-based stimulator device simultaneously to all large whiskers. Each stimulus consisted of 500 μm back-and-forth deflections with 1 ms rise time. One hundred right-sided and 100 left-sided stimuli, alternated within one block, with 9 s inter-stimulus intervals were presented. Recordings were further filtered offline for LFP analyses between 150–250 bandpass and 50 Hz notch filter.

Analyses were performed using Cartool software by Denis Brunet (http://brainmapping.unige.ch/cartool), and Matlab toolboxes (MathWorks). LFPs were calculated offline by averaging responses 100 ms prestimulus to 500 ms poststimulus. One-dimensional current source densities (CSD) were calculated as the product of the second spatial derivative of the electric potential along the cortical depth. The conductivity tensor was assumed to be constant and CSD were then estimated as the finite-difference second spatial derivative of the LFP using the following formula: $CSD = -[V(z + \Delta z, t) - 2 V(z, t) + V(z - \Delta z, t)] / \Delta z^2$; where $V(z, t) = $ measured voltage at subpial depth $z$, $t = $ time, $\Delta z = 100 \, \mu m$[65].

**Retrograde labeling**. To highlight input cells directly connected to layer 2/3 electroporated neurons, we took advantage of rabies-mediated monosynaptic retrograde tracing. Animals were electroporated at E18 with a mix of three plasmids: TVA to restrict the rabies infection only to the layer 2/3 electroporated cells, RabG to allow monosynaptic propagation of the virus, and either dnTCF4 or GFP constructs. At P7, rats were anesthetized with Isofluran in a mix of 30% of oxygen and 70% of air, and were head fixed in a stereotaxic apparatus. A single injection of EnvA pseudotyped and G-deleted rabies virus (EnvA-ΔG-Rabies-mCh, Salk Institute; 150 nl of $1.4 \times 10^7$ TU/μl) was performed in the somatosensory cortex (5 mm lateral and 1.5 mm caudal from the Bregma). 7 days after virus injections (i.e., at P14) we perfused animals, extracted and post-fixed brains, and analyzed 300 μm thick vibratome sections by counting the number of green fluorescent cells (starter cells) in the supragranular layers, and the number of red fluorescent cells (input cells) separately in infragranular, granular and supragranular layers, as determined by Hoechst nuclear staining.

**Cortical slice preparation and time-lapse imaging**. We performed confocal time-lapse imaging of migrating neurons at P1 and axonal growth at P1 and P3 on 300 μm thick acute coronal slices, as previously described[18]. Time-lapse recordings were acquired during 12–14 h (1 image/10 min) on a Nikon A1r upright laser scanning confocal microscope equipped with a 20 × 0.45 CFI ELWD Plan Fluor objective. Video alignment and single-cell tracking were performed with ImageJ Software. The results for speed measurements (migration and axonal growth) come from independent experiments, each having at least 30 tracked cells/axons for each experimental group. To explore the dynamics of TOPdGFP intensity at the single-cell level, we carried out dual color acquisition of time-lapse images of TOPdGFP/RFP co-electroporated cells. The ratio of TOPdGFP/RFP intensity was color-coded and displayed in time-lapse sequences. Prior to acquisition, specific attention was given to gain and laser power settings to avoid signal saturation in each channel. Values of TOPdGFP/RFP ratio at the level of cell soma over time were measured and plotted individually. We then averaged the cellular GFP/RFP ratios of all time points in each of five different bins of the cortex (corresponding to cortical layers as depicted in Fig. 1a). Intensity measurement was done manually using the NIS Elements Software (Advanced Research, Nikon).

**Image acquisition and analysis**. For analyses at high-power magnifications, a Nikon A1r confocal laser scanning microscope was used with 20 × 0.75 CFI Plan Apochromat VC and 60 × 1.4 CFI Plan Apochromat VC objectives and laser illumination diodes 405 nm, 488 nm, 561 nm, and 640 nm. Analyses were done on at least three sections per brain within the dorsolateral barrel subfield identified by Hoechst-stained anatomical and cytoarchitectural landmarks.

For the extent of electroporated regions, entire brains were examined with a Leica M165 FC epifluorescent microscope and photographed with a digital camera (Scion Corporation, CFW-1312C). Images were then compared to surface maps of rat brains by superposition, and the percentage of overlap between the electroporated region and the different cortical areas was calculated.

Quantification of TOPdGFP/RFP or SAREdGFP/RFP double-labeled cells was accomplished on large field, stitched confocal images at P3, P7 and P10 to assess Wnt activity and P10 and P35 to assess neuronal activity. For image capture, the intensity of fluorescent excitation of cells, gain and black levels were kept constant for each session of acquisition. The pixel intensity threshold for GFP was adjusted such that the tissue background corresponded to level 0. Red fluorescent cells displaying GFP fluorescence intensity above this level were considered GFP positive and counted as RFP/GFP double-labeled cell. The total amount of double-labeled cells was normalized by the total number of RFP positive cells. For TOPdGFP/RFP analysis we then divided the proportion of GFP positive cells in the dnTCF4 condition with the same proportion observed in the control brains (i.e., Wnt-activity ratio).

For cell positioning analysis, coronal slices were examined with a Nikon Eclipse 80i epifluorescent microscope and photographed with a digital camera (Retiga EX; Qimaging) controlled by Pictureframe software (MBF Bioscience). Large field images were assembled using Adobe Photoshop. Using MetaMorph software, cell coordinates were projected on the closest curve following the path of radial glia

fiber to estimate the relative percentage of the migration progress. Ventricular and subventricular zone were not included in the analyzed migratory path. Fractions of migrating cells for each 20% of the developing cortex were defined in bins corresponding to cortical layers depicted in Fig. 1a. For each brain, three slices were analyzed and averaged. Cell positioning was also displayed by heatmaps created by the open source statistical software R. In such heatmaps all brains were represented and were color coded according to the proportion of cells present in the above-mentioned five cortical regions.

*Axonal density measurement* was accomplished on large field stitched confocal images at P14. For this purpose 50 μm vibratome coronal sections were entirely imaged. Axonal density was evaluated by average intensity measurement in the S1/S2 border region at the level of layer 2/3 (L2/3), layer 5 A (L5A) and CC. After background subtraction, the intensity of GFP signal in the contralateral CC, L5A and L2/3 was normalized by the ipsilateral L5A intensity signal. Average intensity was measured in standard ROIs centered to the axonal arbors using the NIS Elements Software (Advanced Research, Nikon).

The number of descending axons in the electroporated hemisphere was determined by the number of axons crossing layer 4 normalized by the density of electroporated cells (i.e. number of electroporated cells/length of analyzed segment)

Morphological reconstruction of the dendritic tree of single cells at P14 was done on 50 μm vibratome, coronal sections using the Neurolucida software (MBF Bioscience).

The distance between the cell membrane of migrating cells and radial fiber was measured using Metamorph software. The distance was calculated every 1 μm for the initial segment (40 μm) of the leading process starting from the cell body.

**Acute slice preparation and electrophysiology**. A total of 300 μm thick acute coronal brain slices were cut with a vibratome in cooled cutting solution bubbled with 95% $O_2$ and 5% $CO_2$ containing (in mM): KCL: 2.5; NaHCO₃: 26; NaH₂PO₄: 1.25; MgSO₄: 10; CaCl₂: 0.5, Glucose: 11, Sucrose: 234. Layer 2/3 electroporated cells in acute slices were recorded under an epifluoresence microscope (BX51 WI, Olympus) to allow their detection. Whole-cell patch-clamp recordings were performed in a chamber maintained at 32 °C and continuously perfused at 3 ml/min with artificial cerebrospinal fluid (aCSF) at pH 7.4 bubbled with 95% $O_2$ and 5% $CO_2$ containing (in mM): NaCl: 126; KCL: 2.5; NaHCO₃: 26; NaH₂PO₄: 1.25; MgSO₄: 2; CaCl₂: 2; Glucose: 10. The internal solution contained (in mM) KGluconate: 135; KCl: 4; NaCl: 2; HEPES: 10; EGTA: 4; Mg ATP: 4; NaGTP: 0.3. For miniature EPSCs recordings, neurons were held at −70 mV with TTX (0.5 μM) continuously present in the aCSF in order to block action potentials. For the determination of intrinsic electrophysiological properties, neurons were recorded in current clamp configuration with NBQX (20 μM), AP5 (50 μM) and Gabazine (10 μM) continuously present in the aCSF in order to block the AMPA/Kainate, NMDA, and GABAA receptors, respectively. Signals were amplified by a Multiclamp 700B patch-clamp amplifier (Molecular Device), digitized at 20 kHz and filtered at 4 kHz by a Digidata 1550 (Molecular Device) under the control of pClamp. Data were analyzed using Clampfit 10.4 (Molecular Device).

**Behavioral tests**. Bilaterally electroporated, P35 female and male rats were used for all behavioral experiments. All behavioral tests were performed during the first 6 h of the light phase of the light–dark cycle. For sensory behavior tests, during the habituation period, animals were handled for 7 consecutive days by the experimenter, and in the last 3 days animals were kept in the testing room, and they were placed for 2 min per day on the closed gap-crossing platform (40 × 40 cm elevated platform). During this 3 days animals were also trained to walk in a regular running corridor (1 m long with 1 cm grid distance). Haptic skills and sensorimotor coordination were examined in animals first by the gap-crossing test. Rats were placed on the gap crossing platform, this time with a central 7 cm wide gap (size requiring animals to extend their heads and to use their whiskers to estimate the width of the gap) separating the two sides of the chamber: one brightly illuminated and one obscured and containing nesting material from the home cage. Animals placed to the bright side were let to explore and to cross freely, while measuring the time to cross and the number of whisking-trials before each crosses. According to pre-established exclusion criteria, animals were excluded after failed attempts (i.e. falling in the gap). Next, during the grid-walking test, sensorimotor abilities (limb placement accuracy, coordination) of both hindlimbs and forelimbs were examined by assessing the aptitude of the rats to walk in an identical, but irregular running corridor (long 1 m with 1 to 3 cm grid distance). The step error score was calculated by summing the full (i.e. sliding in the gap between grids, 1 error score) and half foot-faults (i.e. incorrect grip on a grid, 0.5 error score) for both hind and forelimbs. Animals were tested only once on the irregular running corridor. According to pre-established exclusion criteria, animals who did not pass voluntarily through the irregular running corridor were excluded. To assess proprioception and complex sensorimotor functions, we finally used the adhesive patch removal task. A 6-mm diameter circular adhesive patch was placed on the plantar surface of one hindpaw, after which rats were released in the testing arena and observed for 240 s. Latency to detect the first patch (snout contact with the patch) as well as the time taken to remove the patch was measured. Rats underwent three consecutive trials, and values were averaged for each animal. General locomotor activity was examined by the open field test. Rats were placed in a 50 × 50 cm arena in which we monitored and recorded locomotion during 10 min. Videos

were then analyzed using ANYmaze tracking software (Stoelting Co), and the total distance travelled was calculated. To assess social behavior we performed the 3-Chamber Sociability and Social Novelty Test, based on previously described experimental setups[66, 67]. Briefly, animals were tested during 3 session of 10 min (30 min in total) in a three-chambered box with identical openings between the chambers. During the first, habituation session (10 min) animals were free to explore the three-chambered box. Measuring time spent in each side chamber during habituation, we verified that no preference existed for either the right or left side. In the second, sociability session (10 min) we introduced an animal of equal weight and sex (Stranger 1) under a metallic wire pencil cup in the right chamber of the box and an empty, but identical pencil cup in the left one. We tested the preference of the animals for a stranger vs. an object measuring the time spent in each side chamber of the box. We calculated social preference index by subtracting time spent with the Stranger 1 from time spent with object and dividing that with the total time spent with the object and Stranger 1. In the third, social novelty session (10 min) we introduced a second animal (Stranger 2) under the left pencil cup and we again measured the time spent in each compartment. Social novelty index was calculated by subtracting time spent with the Stranger 2 from time spent with Stranger 1 and dividing that with the total time spent with Stranger 1 and 2. During this 30 min test we also measured the total time spent with self-grooming (a common form of repetitive movements in rodents) and sniffing the strangers through the pencil cup (commonly used indicator of social interaction)[66]. Stranger rats were housed in cages distant from the test animals and in separate cages to avoid all physical contact before the experiment, and each stranger was used once per day.

Using the variables generated by behavioral tests we made PCA using the open source statistical program R, in order to plot all animals based on the first two principal components.

**Cell sorting by flow cytometry and RNA extraction**. Six brains per condition per experiment of P1 pups electroporated with either GFP or dnTCF4 were dissected in ice cold aCSF (containing in mM: KCL: 2.5; NaHCO$_3$: 26; NaH$_2$PO$_4$: 1.25; MgSO$_4$: 10; CaCl$_2$: 0.5, Glucose: 11, Sucrose: 234) and cut coronally (400 μm) with vibratome in ice cold aCSF bubbled with 95% O$_2$ and 5% CO$_2$, and containing 3 mM kynurenic acid (Sigma). Positive electroporated slices were selected under a fluorescent stereomicroscope and incubated with oxygenated aCFS/kyn.acid at RT for 1 h. Slices were digested with 0.5 mg/ml pronase (Sigma), then washed with aCFS/kyn.acid. The area of interest was dissected under a stereomicroscope and mechanically dissociated on ice. Fluorescent cells were sorted on the MoFlo Astrios (Beckman) and collected in RNAlater (Sigma) at 4 °C. RNA extraction was performed using RNeasy Micro kit (Qiagen).

**Quantitative real-time PCR analysis**. Quantitative real-time PCR analysis was performed as previously described in ref. [18]. The following primers were used: C-Kit (sense: 5′-GCCAGGAGACGCTGACTATC-3′, antisense: 5′-GGGTAGGCCT CGAACTCAAC-3′); Wnt3A (sense: 5′-CGTCTATGCCATGCGAGCTCA-3′, antisense: 5′-CATCGCCAGTCACATGCACCT-3′); Cyclophilin (sense: 5′-TCAC CATCTCCGACTGTGGA-3′, antisense: 5′-AAATGCCCCGCAAGTCAAAGA-3′). PCR was performed in duplicates for all samples, and the relative expression of target transcripts was calculated using the comparative CT method, and normalized to that of Cyclophilin mRNA.

**Library preparation and RNA-sequencing**. Reverse transcription and pre-amplification of cDNA was achieved using SMARTer Ultra Low RNA kit (Clontech). RNA-sequencing libraries of the collected cDNA were prepared using Nextera XT DNA library prep kit (Illumina). Libraries were multiplexed and sequenced with 100 bp paired-end reads using HiSeq2500 platform (Illumina) with an expected depth of 60M reads, at the iGE3 Genomics Platform of the University of Geneva. The sequenced reads were aligned to the latest reference assembly for rat genome (UCSC-rn5) using the read-mapping algorithm TopHat[68]. The number of reads per transcript was calculated with the open-source HTSeq Python library[69].

**RNA-Sequencing data analysis**. The normalization and differential expression analysis was performed with the R/Bioconductor edgeR package v.3.4.2[70]. All the analyses were computed on the UG Vital-It cluster administered by the Swiss Institute of Bioinformatics. PCA was performed for all transcripts from the collected samples (6 dimensions) and top three PC, representing 82% of total variance were identified. DEGs were identified by the amplitude and statistical significance of the fold change expression level (absolute fold change > 0.9 and false discovery rate (FDR) < 0.1). For each sample, read counts were divided by the total number of mapped reads and multiplied by 10$^6$ to obtain normalized Reads Per Million (RPM) and transformed in log$_2$. The minimum data value in log$_2$ RPM was set to log$_2$10.

In order to identify DEGs related to migration, the publicly available gene ontology database Genego (https://portal.genego.com) was used to make enrichment analysis on migration-related gene ontologies (GOs). We chose gene ontologies containing keywords either "migration" or "locomotion", except for those specified for cell types other than neurons. We then selected all of our DEGs

that were represented in any of the chosen GOs. Using the open source statistical software R, we created a heatmap displaying the normalized RPM transformed in log$_2$ representing the 65 DEGs linked to migration ontologies. In box plot representations, the relative normalized expression levels in log$_2$ of selected genes were displayed for control and dnTCF4 groups.

**Statistical analysis**. No statistics were used to determine group sample size; however, the used sample sizes were similar to those previously published by our group and others. Analysis of experiments was done in a blinded fashion, by tagging animals and naming images or videos of brains during analyses independently from the group they belonged to (control or experimental). All biological replicates (n) are derived from at least three independent experiments. All column graphs are expressed as mean ± s.e.m. The normality of the distribution of data points was verified using D'Agostino & Pearson omnibus normality test. For variables having normal distribution statistical significance was calculated using either two-sided, unpaired Student's t test or ANOVA (one-way or two-way) followed by Bonferroni post-test, as indicated. When normality tests failed to show normal distribution, non-parametric Mann–Whitney or Kruskal–Wallis and Dunn's multiple comparison tests were used, respectively. If variances of compared groups were unequal Welch's correction was used for Student's t tests. All statistical tests were realized with GraphPad Prism 6 Software. Statistical significance was defined at *$P < 0.05$, **$P < 0.01$, ***$P < 0.001$, and ****$P < 0.0001$.

**Data availability**. The data supporting the findings of this study are available within the article and its Supplementary Information files and from the corresponding author on reasonable request.

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

## Acknowledgements

We thank E. Husi and C. Saadi, the Genomics Platform, FACS Core Facility and Bioimaging Core Facility of the University of Geneva for technical assistance; L. Telley for his help with the RNAseq analysis. B. Wehrle-Haller for the C-Kit expression plasmid; D. Muller for the SAREdGFP plasmid. This work was supported by the Swiss National Foundation (grant number: 31003A_159795/1) to J.Z.K.

## Author contributions

R.B., K.E. and J.Z.K.: Conceived, designed the experiments and wrote the paper. R.B. and K.E.: Performed and analyzed the experiments. L.C.-P.: Helped with the data analysis. B.V.: Performed Neurolucida reconstruction. C.Q.: Performed intracortical recording and M.D.R. performed patch clamp experiments. P.S. and S.O.: Designed and constructed the plasmids. M.B.: Helped with the experiments on neuronal migration.

## Additional information

**Competing interests:** The authors declare no competing financial interests.

