## [Peer Review File · Nature Communications]

Reviewers' comments:

Reviewer #1 (Remarks to the Author):

In this manuscript Bocci et al., convincingly demonstrate that Wnt/C-Kit signaling regulates glia-guided radial migration in rat somatosensory cortex. Delays in migration of callosal projection neurons resulting from down-regulation of Wnt signaling disrupts their positioning in layer II/III, retards afferent connectivity, generally lowers neuronal activity, leading to deficits in callosal projections. Rats with these structural deficits demonstrate ASD like behavioral deficits. The authors went on to show that over-expressing the Wnt-downstream effector C-Kit rescues abnormal interhemispheric connections as well as behavioral alterations. Furthermore, callosal connectivity defects can be rescued by chemogenetic activation of migration delayed callosal projection neurons. Overall, this is a fairly well executed study demonstrating how transient delays in neuronal migration affects cortical circuit formation and behavior. Delineation of mechanistic underpinnings of this process is a major strength of this work. The potential high significance of this work for cortical development field and neurodevelopmental disorders makes this manuscript appropriate for publication in Nature Communications.

However, there are several major and minor concerns that need to be addressed:

- 1) Please forthrightly discuss the differences in cerebral cortical phenotypes observed following embryonic inactivation or activation of b-catenin in mice. Addressing the comparative differences between this study and earlier work on b-catenin signaling/cortical development will help strengthen this work.
- 2) Please address how the authors controlled for dnTCF4 effect on radial glia.- especially in studies related to neuron- glial adhesion.
- 3) Please add complete details on the extent of electroporated neurons in somatosensory cortex (and in other cortical areas)- especially in rats subjected to behavioral studies.
- 4) The following end to first paragraph in pg.9 is confusing "Thus, the reduced callosal connections of delayed neurons were not due to abnormal axonogenesis or axonal guidance, but instead were rather related to decreased contralateral branching and arborization. Importantly, this axonal defect could not be directly attributed to perturbed Wnt signaling, since Wnt activity returned to control levels by the time of arborization (i.e. P7-P14; Supplementary Fig. 5a), and Wnt LOF induced after migration (P3-P7) did not decrease axonal arbor development (Supplementary Fig. 5b). Thus, the late arrival of CPNs could lead by itself to deficient axonal arborization." Describe the axonal phenotype of delayed migration callosal neurons simply and directly.
- 5) Did rats in which DREADD (hM3Dq) was activated in late-arriving CPNs also display behavioral rescue?

Reviewer #2 (Remarks to the Author):

This is a fascinating and impressive story. I find the data largely fairly compelling and the overall conclusions to be pretty well supported. I do have a few comments that I think require some careful consideration by the authors and perhaps some modulation of the way some of the conclusions are framed about the data.

The authors show nicely that over-expression of C-Kit rescues the migration delay and many of the other abnormalities but they never give due consideration whether over-expression of C-Kit has a phenotype on its own. What does C-Kit over-expression do to wt animals? If it doesn't accelerate

migration beyond normal speeds can the authors provide some ideas as to why that would be. Is it ligand limited? What about providing C-Kit ligand excess in normal rats, does that speed up migration? These are simple experiments and they would close the loop about the relationship with Wnt signaling.

In addition, the authors argue that it is the delay in migration that leads to the connectivity problems, not the defects in Wnt or C-Kit signaling. However, they aren't really able to conclude this in my mind. Since they have no readout for C-Kit signaling there could be long lasting changes in signaling downstream of this receptor. In addition, the Top-GFP is not really a sensitive enough way to examine Wnt signaling as there may be a variety of non-canonical or subtle canonical changes that are more long lasting. I think it is an overstatement to conclude for sure that the migration delay is the critical factor rather than the signaling changes. One way to firm this up, if the authors care to hammer this point home, would be to devise a different way to delay migration that has nothing to do with Wnt or C-Kit signaling and show similar abnormalities.

These two criticisms aside, I think this is quite a nice piece of work and will help us to understand this system better.

Reviewer #3 (Remarks to the Author):

In the manuscript "Delayed migration of callosal projection neurons alters interhemispheric connections, somatosensory functions and social behavior", Bocchi et al . provided evidence for a role of Wnt/C-Kit signaling in regulating glia-guided radial migration in rat somatosensory cortex. They further associated transient delay of migration of L2/3 neurons with impaired interhemispheric connectivity and abnormal social behavior. The results cover much ground, use many state-of-the-art techniques and provide several interesting leading. However, there are a number of leaps in logic in the manuscript that make it hard to connect the dots in a meaningful fashion.

Major comments

1. In investigating the role of Wnt/beta-catenin signaling in neuronal migration, the authors used TOPdGFP, dnTCF4 and (δ)Dvl2. TOPdGFP is a fluorescent reporter based on TOPFLASH, originally designed by the Hans Clevers lab. In luciferase assays, the results of TOPFLASH are ratioed over that of FOPFLASH, which contains mutations in the Wnt/beta-catenin transcriptional target sites. Here, no equivalent controls are performed. I understand that in a previous paper studying the role of Wnt/beta-catenin signaling in neuronal migration (Biotard et al., Cell Reports, 2015), the authors showed that the fluorescence level of TOPdGFP can be modulated by Wnts. However, they did not show complete loss of the reporter signal under any Wnt manipulation. Thus, it is not clear if the changes in GFP fluorescence reflect only changes in Wnt signaling, or also changes in other unrelated signaling pathways. Results from a FOPdGFP reporter, or equivalent control need to be provided to address the specificity issue. This is important because here the absolute fluorescence of TOPdFGP is interpreted.

2. Related to the first point, because of the complexity of the Wnt/beta-catenin signaling pathway, with its canonical and multiple non-canonical alternatives, one needs to manipulate both Wnt and beta-catenin in order to make a statement about this pathway. To confuse the issue further, Tcf4 has other homologs including Lef1, Tcf3, and Tvcf7 and Dvl2 has homologs Dv1 and Dvl3. Thus the dominant negative constructs can have very complicated effects, and furthermore, full-length constructs are not included as controls. The beta-catenin conditional knockout mouse is publically

available and its RNAi sequences are also published.

3. The effects of C-Kit on neuronal migration and axon formation as reported by Guijarro et al. (Dev Neurobiol. 2013) are similar, but significantly different from that reported by the authors. Thus, the authors need to demonstrate that the effects of C-Kit RNAi are similar to that of dnTCF4 in their system. Also, to identify C-Kit as a downstream effector of Wnt/beta-catenin dependent transcription, the Wnt/beta-catenin responsive sites in the C-Kit promoter need to be identified. Minimally, additional data demonstrating that C-Kit is a transcriptional target of Wnt/beta-catenin signaling need to be provided by qPCR and Western blots.

4. Related to 3, the authors should analyze changes in the expression in known Wnt targets in their microarray data to provide further evidence that their dnTCF4 construct indeed reduced Wnt/beta-catenin dependent transcription.

5. The changes in PSD95 density presented in Fig 5 are very striking. Since the authors can perform slice physiology experiments, mEPSCs results would provide significant support for changes in synapse density.

6. Since the authors demonstrated that elevated neural activity can rescue the effects of dnTCF4 on neuronal migration, they should also perform the same rescue for the social behavior experiments.

7. Western blots and immunohistochemistry data need to be provided for verification of the constructs used.

Reviewer #4 (Remarks to the Author):

Bocchi et al. present a relevant and novel study that thoroughly shows far-reaching effects of Wnt pathway manipulation on L2/3 somatosensory neurons. The authors use a range of techniques to nicely demonstrate potential mechanistic causes of defects in migration upon manipulation of the Wnt pathway, as well as changes in callosal development, cortical activity and behaviour that result from these changes. Convincing evidence is presented that decreased Wnt signalling effects the stability of the leading process of migrating neurons and their attachment to radial glia. Crucially, many of these deficits are eliminated upon rescue of the Wnt pathway deficits. This very nice paper is of high quality, relevance and impact for the field. There are some points that require clarification/ adjustment before the paper is published.

Major point:

1) The conclusion stated throughout the manuscript is that the late arrival of migrating neurons causes the cascade of other defects. This is a plausible conclusion and is supported by the data, but, at present, the evidence indicates that early TCF4/Wnt/Ckit signalling is required for both neuronal migration and synapse formation given the following:

-The authors show decreased afferent synaptic inputs from layer 4 neurons on this ipsilateral side using pseudorabies virus. It is likely that the decreased synaptic inputs could cause defects in contralateral axonal arborization.

-The data showing that increased activity using DREAD's can rescue the axonal arborization phenotype supports that rescuing synaptic development is possible but this may be independent of delayed migration.

-In addition, the authors have done a good job in trying to address this in supplementary figure 5 (which is important data and should be part of the main figures). In this figure the authors delay down regulation of Wnt signalling until P3-P7 and do not see an effect on axonal arborization. In addition Wnt signalling is back to normal levels during the time of axonal arborization. They conclude that therefore the defect is not due to Wnt signalling per se but due to delayed migration. However, it could also be due to the early effects of delayed synaptic input.

The conclusions are currently very strong and should be modified to include additional possibilities. At present, the data supports that both delayed cell migration and defects in synaptic input are correlated with axonal arborization and behavioural deficits.

Minor points:

N values and statistics:

2) The n numbers stated in figure legends vary between "brains" "animals" "brain sections" "slices" and "cells". At times, two figure panels that are similar in methodology have n numbers presented as brains and cells respectively (e.g. Fig. 2A and B). The authors should use animals as the n number, and provide convincing justification if using slices, experiments or cells, and include a clear definition of what each of those terms means and how many animals were included in the collection of each for statistical analyses.

3) The Sink ampl. Bar graph in Fig. 4a was statistically compared using a t-test, but the individual data points look to be not normally distributed (skewed). The authors should ensure that tests of normality were performed on all data sets.

Other points

4) It is the convention in the field to use "Upper" layers and "Deeper" layers of the cortex rather than "high", "mid" and "low" CP for early stages of cortical development (before E18 in rat), but for later ages the actual cortical layer number (I-VI) should be used. For example, "high" CP appears to correlate with layer II/III, therefore layer II/III should be used. The authors could use a nuclear marker such as DAPI or a Nissl stain to help delineate the different cortical layers. Not using the conventional names made it hard to understand at first. In addition, because the data is regarding low and high levels of Wnt activity this is current nomenclature was confusing.

5) Figure labelling: rather than using left and middle for different panels in (for example Fig 2) use b and b'. This should be applied to all figures.

6) Why are Fig. 1e (middle panel) and Fig. 3f (right panel) not presented in the same way as Fig. 1d (right panel)? These are important panels, as they show the transience of the migration delay upon dnTCF4 administration and the rescue of the defects with C-Kit. These findings are crucial to the impact of the paper and would greatly benefit from statistical analysis, similar to that performed in Fig. 1d (right panel). Similarly, a whole cortex image of a control brain at P7 would be useful to compare to Fig 1e (left panel), as the provided image looks similar to the control at P3, but this is not an appropriate comparison.

7) Are the right panels of Fig. 1a single neurons or averages? This should be made clear in the figure legend.

8) Figure 3, panel B does not add much information to the figure that is not already in the text.

9) Is it appropriate to normalize contralateral callosal axon branching by ipsilateral branching? Is there evidence that the ipsilateral branching is not increased/decreased by experimental manipulations? The authors provide evidence that activity is changed in the ipsilateral side and there is a reduction in synaptic contacts from L4 (Fig. 5), so it would not be unreasonable to hypothesise that ipsilateral axonal branching may also be affected.

10) At present there is no evidence showing that dnTCF4 or Δ DVL2 alter the Wnt signaling pathway (apart from tangentially with the rescue experiments). This is an important basis to the conclusions of the experiments, and therefore the instances of "data not shown" on page 5 and 6 describing

Luciferase assays/electroporation experiments that have been done to show this should be included in the supplementary figures.

11) The references in text and in figure legends to Supp Fig 4 and Fig 5 become confusing with the terms "ipsilateral" and "contralateral" used in reference to trimmed whiskers as well as site of electroporation (which can be opposite sides). This should be clarified, perhaps with the aid of diagrams. Also, changes in activity are found ipsilaterally (on the side of electroporation) with eSARE, but not with intracortical recordings (LFP). Do the authors have an explanation for this? Why are the eSARE experiments (Supp Fig. 4b and Fig 5a) on the two cortical hemispheres performed under different conditions? (enriched environment vs. not).

12) The manuscript and figures would benefit from editing to amend typographical errors and improve the clarity of writing.

Overall, this is a very nice study.

Reviewers' comments:

Reviewer #2 (Remarks to the Author):

The authors have done a nice job of responding to most of the concerns.

However, I still believe it is difficult to know whether it is the migration "delay" that results in the phenotype or whether it is separate effects on activity caused by changes in Wnt signaling. The authors persist in making this overblown conclusion. They have modulated it in some places but it is still obvious that this is their bias without adequate proof (eg see the title).

The phenotype described previously with the c-kit overexpression in another study sped up migration (as the authors discuss in the revised manuscript and acknowledge in the rebuttal). It seems that the authors really need to reproduce this and determine whether it has the opposite effects of their Wnt manipulations.

In summary, I believe the authors are still overstating the meaning of their data - it is unclear whether it is the delay in migration or the effects on Wnt signaling that are the key factors in regulating the phenotype. Until they find another way to delay migration not dependent on Wnt signaling then they can't come to the conclusions that they do in a way that is adequately convincing in my mind.

Reviewer #3 (Remarks to the Author):

The authors have made a significant effort to address all my concerns from my first review. I think the results presented are interesting and would be of interest to the readers of Nature Communications. I only have a few minor suggestions for changes:

1. The results showing that changes in neuronal migration affect callosal projections and mouse behavior were most extensively demonstrated with dnTCF. I understand that the authors provided substantial evidence that this is through the function of TCF in canonical Wnt signaling. However, in an overexpression system there are alternative explanations, and the results may be specific to a weak suppression of nuclear Wnt signaling without affecting cell adhesion at a specific developmental stage and in specific neurons. I would suggest a little more emphasis on dnTCF and TCF. For example, P5, 3 lines from bottom, change to "dnTCF did not seem to affect the radial glia structure." This comment does not affect the novelty of the manuscript.

2. On P10, line 16, an accurate description is: "...that delayed neurons had reduced mEPSC frequency" or "the delayed neurons had longer inter-event intervals". In the associated Fig. 5c, should be "mEPSC amplitude" and "mEPSC inter-event interval".

Reviewer #4 (Remarks to the Author):

The authors have fully addressed the previous comments. The manuscript is greatly improved.

REVIEWERS' COMMENTS:

Reviewer #2 (Remarks to the Author):

The authors have addressed my residual concerns.

Reviewer 1

1) Comment: « Overall, this is a fairly well executed study demonstrating how transient delays in neuronal migration affects cortical circuit formation and behavior. Delineation of mechanistic underpinnings of this process is a major strength of this work. The potential high significance of this work for cortical development field and neurodevelopmental disorders makes this manuscript appropriate for publication in Nature Communications. »

Response: We thank the Reviewer for the positive note.

2) Comment: « Please forthrightly discuss the differences in cerebral cortical phenotypes observed following embryonic inactivation or activation of b-catenin in mice. Addressing the comparative differences between this study and earlier work on b-catenin signaling/cortical development will help strengthen this work. »

Response: We thank the reviewer for this pertinent suggestion. A comparison of the relevant literature on β -catenin signaling in cortical development and our findings is now included in the revised version in the discussion (2nd paragraph).

3) Comment: « Please address how the authors controlled for dnTCF4 effect on radial glia, especially in studies related to neuron-glia adhesion. »

Response: We agree that this point needs clarification. In principle, we cannot exclude the possibility that our *in utero* electroporation procedure targets radial glia as well as migrating neurons. However,

in a large series of experiments, we never found evidence that our plasmids affect the radial glia population. We can detect plasmid-derived fluorescence in radial glia fibers 24h after electroporation, but not 48h or 72h after electroporation when we activate our inducible plasmids. In our opinion, this is due to the fact that electroporated plasmids are not incorporated into the genome and repeated cell divisions “dilute” plasmid concentration until complete disappearance. This effect is particularly strong in the ventricular proliferative zone where during active neurogenesis, such as at E18, radial glia cells go through several rounds of cell division. To further illustrate this point, we added a new Nestin immunofluorescence image in Supplementary Fig. 1d highlighting the similarities of the radial glia organization in control and dnTCF4 electroporated brains.

4) Comment: « Please add complete details on the extent of electroporated neurons in somatosensory cortex (and in other cortical areas), especially in rats subjected to behavioral studies. »

Response: We thank the Reviewer for this pertinent suggestion. Indeed, it is important to determine the topography of affected regions for the adequate interpretation of the results of the behavior tests and this point has not been sufficiently documented in the original manuscript. Using 5mm diameter electroporation paddles on a fetal brain does not allow selective targeting of small cortical areas. However, by controlling the head-paddle angle, we could focus the electroporation fairly well on the primary somatosensory cortex (S1). The regional distribution was controlled meticulously on each brain from animals used for behavior studies by visualizing the internal fluorescence of electroporated regions on whole brains before sectioning (see new Supplementary Fig. 7a, left). Importantly, while in a few animals we observed some labeling in primary visual (V1) and auditory (A1) cortices, motor areas (M1, M2) seemed barely affected in the majority cases. Taken together, these newly included data support the hypothesis that our electroporations affected mainly the somatosensory cortex and adjacent motor areas were not impacted.

5) Comment: « The following end to first paragraph in pg.9 is confusing “Thus, the reduced callosal connections of delayed neurons were not due to abnormal axonogenesis or axonal guidance, but instead were rather related to decreased contralateral branching and arborization. Importantly, this axonal defect could not be directly attributed to perturbed Wnt signaling, since Wnt activity returned to control levels by the time of arborization (i.e. P7-P14; Supplementary Fig. 5a), and Wnt LOF induced after migration (P3-P7) did not decrease axonal arbor development (Supplementary Fig. 5b). Thus, the late arrival of CPNs could lead by itself to deficient axonal arborization.” Describe the axonal phenotype of delayed migration callosal neurons simply and directly. »

Response: Thank you for this observation. We agree that the original text was not sufficiently clear. We have provided a new description in the revised manuscript: Results, last paragraph of chapter “Late arriving CPNs develop altered callosal projections”.

6) Comment: « Did rats in which DREADD (hM3Dq) was activated in late-arriving CPNs also display behavioral rescue? »

Response: We agree with the Referee that this is a crucial issue that should be addressed in this study. Therefore, we have carried out new series of experiments, in which we have co-electroporated the hM3Dq and the dnTCF4 constructs, and activated callosal neurons exclusively during axonal arborization (i.e. CNO injections were finished at least two weeks before the behavior tests). Social behavior was tested as before and the results directly answer the referee’s concern: by restoring post-natal neuronal activity we have also restored social variables to control levels (see new Supplementary Fig 7c and d).

Reviewer 2

1) Comment: « This is a fascinating and impressive story. I find the data largely fairly compelling and the overall conclusions to be pretty well supported. »

Response: We thank the reviewer for the positive remark.

2) Comment: « The authors show nicely that over-expression of C-Kit rescues the migration delay and many of the other abnormalities but they never give due consideration whether over-expression of C-Kit has a phenotype on its own.»

Response: This is a good point. Accordingly, we have considered to carry out C-Kit gain-of-function experiments, however we found the relevant data in the literature. In the paper of Guijarro et al., (2013) the authors used an animal model very similar to the one used in our studies (in utero electroporation of layer 2/3 in rat embryos). They showed that layer 2/3 neurons over-expressing C-Kit arrive earlier to their final cortical positions. The authors concluded that C-Kit accelerates radial migration. This has further been confirmed in culture experiments in which kit-ligand stem cell factor (SCF) enhanced neuronal migration. These results have been cited in the revised discussion (paragraph 3).

3) Comment: « I think it is an overstatement to conclude for sure that the migration delay is the critical factor rather than the signaling changes. »

Response: We agree, this point is very relevant. Although, we are convinced that the proposed cascade of “migration delay/reduced afferent inputs/decreased neuronal activity during critical period development/connectivity aberration” is the most likely scenario, we cannot completely exclude the possibility that long-term changes after a transient decrease of Wnt/C-Kit signaling could also contribute to the observed alteration in callosal axon arborization. We agree with the Referee that the most elegant way to prove the role of migration delay in circuit development would be to find an alternative means to delay migration. Unfortunately, we could not find any useful information in this sense in the literature and were unable to work out a new model during the relatively short period we disposed to revise our paper. Therefore, in line with the Referee’s arguments, we have decided to modify our conclusions in the revised discussion (mainly in paragraph 5, and see also paragraph 1, abstract, etc.).

Reviewer 3

1) Comment: « The results cover much ground, use many state-of-the-art techniques and provide several interesting leading. However, there are a number of leaps in logic in the manuscript that make it hard to connect the dots in a meaningful fashion. »

Response: We thank the reviewer for the positive remark, and we hope that the modifications to the manuscript including several new series of experimental data will help to fill the logical gaps observed by the reviewer.

2) Comment: « In investigating the role of Wnt/beta-catenin signaling in neuronal migration, the authors used TOPdGFP, dnTCF4 and (delta)Dvl2. TOPdGFP is a fluorescent reporter based on TOPFLASH, originally designed by the Hans Clevers lab. In luciferase assays, the results of TOPFLASH are ratioed over that of FOPFLASH, which contains mutations in the Wnt/beta-catenin transcriptional target sites. Here, no equivalent controls are performed. I understand that in a previous

paper studying the role of Wnt/beta-catenin signaling in neuronal migration (Biotard et al., Cell Reports, 2015), the authors showed that the fluorescence level of TOPdGFP can be modulated by Wnts. However, they did not show complete loss of the reporter signal under any Wnt manipulation. Thus, it is not clear if the changes in GFP fluorescence reflect only changes in Wnt signaling, or also changes in other unrelated signaling pathways. Results from a FOPdGFP reporter, or equivalent control need to be provided to address the specificity issue. This is important because here the absolute fluorescence of TOPdGFP is interpreted. »

Response: We agree, the specificity of our manipulations on Wnt signaling is an important issue. Given that the dnTCF4 and Δ DVL2 plasmids have both been described previously (van de Wetering et al., 2002; Korinek et al., 1997 and Schwarz-Romond et al., 2007), we have not included their validation by Luciferase assay using the negative control mentioned by the reviewer (FOP) in the first version of the manuscript. Nevertheless, the validation experiments have been performed and the results are now presented in the manuscript (see new Supplementary Fig. 1c). In this figure, the TOP condition reflects endogenous signaling levels in HEK293T transfected cells, whereas the FOP condition represents the basal non-specific transcriptional activation. Note, that for visualizing the effect of upstream Δ DVL2 on Luciferase activity we added an activating ligand (Wnt1). Importantly, dnTCF4 achieved a greater loss of Luciferase activity than FOP alone. Moreover, in an additional control experiment, using co-electroporation of either dnTCF4 or non-degradable Δ 45 β -catenin with TOPdGFP, we now demonstrate in vivo the modulation of the TOP promoter activity in both directions (see new Supplementary Fig. 1a). In our view, these new data indicate that dnTCF4 specifically inhibits canonical Wnt-signaling related transcription, and implies that in vivo changes in GFP fluorescence depicted in Fig.1a and Supplementary movie 1, reflect real changes in canonical Wnt-signaling.

3) Comment: « Related to the first point, because of the complexity of the Wnt/beta-catenin signaling pathway, with its canonical and multiple non-canonical alternatives, one needs to manipulate both Wnt and beta-catenin in order to make a statement about this pathway. To confuse the issue further, Tcf4 has other homologs including Lef1, Tcf3, and Tvcf7 and Dvl2 has homologs Dvl1 and Dvl3. Thus the dominant negative constructs can have very complicated effects, and furthermore, full-length constructs are not included as controls. The beta-catenin conditional knockout mouse is publically available and its RNAi sequences are also published. »

Response: We agree with the reviewer; Wnt signaling is a very complex process where Wnt ligands differentially regulate canonical and non-canonical signaling pathways and dominant negative constructs could indeed lead to complicated effects. In these studies, we used a limited number of dominant negative constructs, addressed only a fraction of issues pertaining to the canonical pathway and there is much still to learn about the role of Wnt-related pathways in neuronal migration. Nonetheless, we believe that three lines of evidence in the current study strongly support the hypothesis that the Wnt/ β -catenin signaling pathways takes part of a core signaling that regulates the pace of pyramidal cell migration. First, in order to address the concern of the Referee, we have carried new series of experiments using an shWnt3A construct to transiently knock down the canonical Wnt3A ligand expression in migrating cells. We found that Wnt3A knockdown results in migration delay (Supplementary Fig. 1j). This implies that canonical signaling implicated in regulating neuronal migration could be triggered, at least partly, cell-autonomously. Second, we demonstrate that loss-of-function of the dishevelled protein, a cytoplasmic mediator necessary to block the assembly of the β -catenin destruction complex in response to Wnt-ligand binding also leads to migration delay. Finally, loss-of-function of a downstream effector directly responsible for β -catenin mediated transcriptional effects (TCF4) also results in altered migration. These points have been addressed in the first chapter of the Results.

4) Comment: « The effects of C-Kit on neuronal migration and axon formation as reported by Guijarro et al. (Dev Neurobiol. 2013) are similar, but significantly different from that reported by the authors.

Thus, the authors need to demonstrate that the effects of C-Kit RNAi are similar to that of dnTCF4 in their system. Also, to identify C-Kit as a downstream effector of Wnt/beta-catenin dependent transcription, the Wnt/beta-catenin responsive sites in the C-Kit promoter need to be identified. Minimally, additional data demonstrating that C-Kit is a transcriptional target of Wnt/beta-catenin signaling need to be provided by qPCR and Western blots. »

Response: We accept the point raised by the Referee and we have identified Wnt/ β -catenin responsive sites within C-Kit promoter, as suggested, (see new Supplementary Fig. 3f). Moreover, we have performed qRT-PCR studies on P0 sorted, electroporated neurons, which showed a decrease in C-Kit expression after dnTCF4 electroporation (see new Fig. 3c), thereby confirming our RNA sequencing data.

We also agree that differences exist between the study of Guijarro et al. and our work. In our view, the most important difference between the two studies relies in the fact that while Guijarro et al. used constitutive over-expression/knock-down of C-Kit, effective during radial migration as well as during axonal extension, here we elicited merely a transitional perturbation of Wnt-C-Kit signaling, specifically during the migration period. Based on their observations, Guijarro et al. inferred that the correct dosage of C-kit expression in developing pyramidal neurons is required for proper radial migration and callosal axon extension. However, their in vivo experimental setup did not allow them to distinguish between the effects of delayed migration and a direct role of C-Kit in axonal extension. Our results clearly indicate that the axonal arborization deficit is a consequence of the delayed migration and that overexpression of C-Kit specifically during migration and not during the period of axonal growth, rescues normal connectivity. It is of interest that using cultured cortical neurons, Guijarro et al. could not observe any effect after applying the C-Kit ligand, stem cell factor, on cortical axonal extension. It appears hence that our results and conclusions are compatible with the observations of Guijarro et al.

5) Comment: « Related to 3, the authors should analyze changes in the expression in known Wnt targets in their microarray data to provide further evidence that their dTCF4 construct indeed reduced Wnt/beta-catenin dependent transcription. »

Response: We thank the suggestion of the reviewer. The known Wnt targets we have verified in our RNA sequencing data (Lef1, Met, Vcan, Fst, see new Supplementary Fig. 3b) have all shown to be down regulated by dnTCF4.

6) Comment: « The changes in PSD95 density presented in Fig 5 are very striking. Since the authors can perform slice physiology experiments, mEPSCs results would provide significant support for changes in synapse density. »

Response: We thank the suggestion of the reviewer. To address this issue, we have performed additional experiments and integrated mEPSC recording results to further support the observed changes in synapse density (see new Fig. 5c).

7) Comment: « Since the authors demonstrated that elevated neural activity can rescue the effects of dnTCF4 on neuronal migration, they should also perform the same rescue for the social behavior experiments ».

Response: We thank the reviewer for the suggestion; we have performed a new set of experiments, in which we have co-electroporated the hM3Dq and the dnTCF4 constructs, and activated callosal neurons exclusively during axonal arborization (i.e. CNO injections were finished at least two weeks before the behavior tests). Social behavior was tested as before. We found that by restoring post-natal

neuronal activity we could also restore social behavior to control levels (see new Supplementary Fig. 7c and d).

8) Comment: « Western blots and immunohistochemistry data need to be provided for verification of the constructs used. »

Response: We agree with the reviewer that the validation of the constructs should be presented in the manuscript. The validation of our main constructs (dnTCF4, Δ DVL2) was performed by in vitro Luciferase assay and in vivo co-electroporations with TOPdGFP (see also Response to Comment 2).

Reviewer 4

1) Comment: « Bocchi et al. present a relevant and novel study that thoroughly shows far-reaching effects of Wnt pathway manipulation on L2/3 somatosensory neurons. ... This very nice paper is of high quality, relevance and impact for the field. »

Response: We thank the reviewer for the positive remark and for the generally positive opinion on the manuscript.

2) Comment: « The conclusion stated throughout the manuscript is that the late arrival of migrating neurons causes the cascade of other defects. This is a plausible conclusion and is supported by the data, but, at present, the evidence indicates that early TCF4/Wnt/Ckit signalling is required for both neuronal migration and synapse formation given the following:

-The authors show decreased afferent synaptic inputs from layer 4 neurons on this ipsilateral side using pseudorabies virus. It is likely that the decreased synaptic inputs could cause defects in contralateral axonal arborization.

-The data showing that increased activity using DREAD's can rescue the axonal arborization phenotype supports that rescuing synaptic development is possible but this may be independent of delayed migration.

-In addition, the authors have done a good job in trying to address this in supplementary figure 5 (which is important data and should be part of the main figures). In this figure the authors delay down regulation of Wnt signalling until P3-P7 and do not see an effect on axonal arborization. In addition Wnt signalling is back to normal levels during the time of axonal arborization. They conclude that therefore the defect is not due to Wnt signalling per se but due to delayed migration. However, it could also be due to the early effects of delayed synaptic input.

The conclusions are currently very strong and should be modified to include additional possibilities. At present, the data supports that both delayed cell migration and defects in synaptic input are correlated with axonal arborization and behavioural deficits. »

Response: We taken the Referee's suggestion and we have revised our conclusions (see also response to the Referee 2). We also agree with the Referee that early effects of reduced synaptic input could very well be responsible for the altered callosal arborization in the contralateral hemisphere. We believe however that the delayed synaptic input is due to the delayed arrival of cells in layer 2/3. As we speculate in the discussion: "...late arrival generates competitive disadvantage to recruit afferent inputs. Supporting this hypothesis, we identified a reduction in short-range afferent connections, suggesting the perturbation of synaptic integration and subsequent activation of delayed neurons."

3) Comment: « The n numbers stated in figure legends vary between "brains" "animals" "brain sections" "slices" and "cells". At times, two figure panels that are similar in methodology have n

numbers presented as brains and cells respectively (e.g. Fig. 2A and B). The authors should use animals as the n number, and provide convincing justification if using slices, experiments or cells, and include a clear definition of what each of those terms means and how many animals were included in the collection of each for statistical analyses. »

Response: We thank the reviewer for this observation. We have now corrected and unified wording in the legends concerning the n numbers using « brains » whenever it was possible. Where we could not change the original wording (i.e. because n = cells and not brains, see. Fig. 2a) we have added further explanations to the legends. Concerning Fig. 2a and b, please note the differences in the methods used: Fig. 2a depicts images from time-lapse videos and graphs of the leading process length changes in time in individual cells underneath, and besides the column statistics represent the length of leading processes of individual cells, analyzed within post hoc images (n = cells analyzed within post hoc images). Fig. 2b depicts two examples of inverting cells on time lapse videos, while the column statistics represent the proportion of inverted cells within brains (n = brains analyzed via post hoc coronal sections). This difference explains the difference in the n numbers.

4) Comment: « The Sink ampl. Bar graph in Fig. 4a was statistically compared using a t-test, but the individual data points look to be not normally distributed (skewed). The authors should ensure that tests of normality were performed on all data sets. »

Response: We thank the reviewer for this comment, we have re-tested normality and indeed, using the D'Agostino-Pearson test, normality was not confirmed on the dataset of Fig. 4a. We have changed statistics accordingly (instead of t test, parametric test we now use Wilcoxon, non-parametric test).

5) Comment: « It is the convention in the field to use “Upper” layers and “Deeper” layers of the cortex rather than “high”, “mid” and “low” CP for early stages of cortical development (before E18 in rat), but for later ages the actual cortical layer number (I-VI) should be used. For example, “high” CP appears to correlate with layer II/III, therefore layer II/III should be used. The authors could use a nuclear marker such as DAPI or a Nissl stain to help delineate the different cortical layers. Not using the conventional names made it hard to understand at first. In addition, because the data is regarding low and high levels of Wnt activity this is current nomenclature was confusing. »

Response: We have taken the the reviewer's suggestion. In the revised version of the manuscript we have changed the original « high », « mid » and « low » terms to layer numbers. As suggested, for the layer discrimination we used the DAPI channel on fluorescence images. Additionally, we used « bin » with a corresponding number to label artificially designated layers (Fig 1a, d and e).

6) Comment: « Figure labelling: rather than using left and middle for different panels in (for example Fig 2) use b and b'. This should be applied to all figures. »

Response: We agree with the reviewer that descriptions in some figure legends in the original manuscript were complicated. To facilitate navigation between the legends and the figures, we have now applied individual numbers to the sub-panels whenever it was necessary and possible. To further simplify without perturbing comprehension on simple multi-panels we have erased « left » and « right » designations (e.g. image and graph within one panel, and from the text it is unequivocally understandable that we refer either to the graph or the image).

7) Comment:« Why are Fig. 1e (middle panel) and Fig. 3f (right panel) not presented in the same way as Fig. 1d (right panel)? These are important panels, as they show the transience of the migration delay upon dnTCF4 administration and the rescue of the defects with C-Kit. These findings are crucial to the

impact of the paper and would greatly benefit from statistical analysis, similar to that performed in Fig.1d (right panel). Similarly, a whole cortex image of a control brain at P7 would be useful to compare to Fig 1e (left panel), as the provided image looks similar to the control at P3, but this is not an appropriate comparison. »

Response: The reason for the altered presentation was purely esthetic, however, we agree with reviewer that a uniform presentation would help understanding of the data. In line with that, we have added the statistics similar to the one performed in Fig. 1d to Fig. 1e and the Fig. 3f, as well (presented in Supplementary Fig. 1e and Supplementary Fig. 3g, respectively). Additionally, as requested, we have inserted a whole cortex control image to Fig. 1e.

8) Comment: « Are the right panels of Fig. 1a single neurons or averages? This should be made clear in the figure legend. »

Response: The panels in question of Fig. 1a represent the single neurons depicted in the time lapse videos (See Supplementary Movie 1), the extracted images of which are presented in the same panel. We have clarified this detail in the figure legends.

9) Comment: « Figure 3, panel B does not add much information to the figure that is not already in the text. »

Response: We agree with the reviewer, and we removed this panel.

10) Comment: « Is it appropriate to normalize contralateral callosal axon branching by ipsilateral branching? Is there evidence that the ipsilateral branching is not increased/decreased by experimental manipulations? The authors provide evidence that activity is changed in the ipsilateral side and there is a reduction in synaptic contacts from L4 (Fig. 5), so it would not be unreasonable to hypothesise that ipsilateral axonal branching may also be affected. »

Response: We thank the Referee for this very pertinent question. We have never seen striking differences in the ipsilateral axonal branches. This is now confirmed by our new analyses presented in Supplementary Fig. 4d. Here, we have measured intensity in ipsilateral Layer 5 and normalized it to the density of electroporated cells (similarly to the normalization of descending axons in Supplementary Fig. 4e).

11) Comment: « At present there is no evidence showing that dnTCF4 or Δ DVL2 alter the Wnt signaling pathway (apart from tangentially with the rescue experiments). This is an important basis to the conclusions of the experiments, and therefore the instances of “data not shown” on page 5 and 6 describing luciferase assays/electroporation experiments that have been done to show this should be included in the supplementary figures. »

Response: We agree with the reviewer, and have now included the validation of the constructs by the Luciferase assay in the new Supplementary Fig. 1c. For in vivo validation, a co-electroporation of dnTCF4 and Δ 45 β -catenin with TOPdGFP is now presented in the new Supplementary Fig. 1a. (For more details, please see also response for Reviewer 3, Comment 2).

12) Comment: « The references in text and in figure legends to Supp Fig 4 and Fig 5 become confusing with the terms “ipsilateral” and “contralateral” used in reference to trimmed whiskers as well as site of electroporation (which can be opposite sides). This should be clarified, perhaps with the

aid of diagrams. Also, changes in activity are found ipsilaterally (on the side of electroporation) with eSARE, but not with intracortical recordings (LFP). Do the authors have an explanation for this? Why are the eSARE experiments (Supp Fig. 4b and Fig 5a) on the two cortical hemispheres performed under different conditions? (enriched environment vs. not). »

Response: We accept the criticism. The corresponding text has been revised and now we use contralateral and ipsilateral designation relative to the electroporation side, as can be read in Results chapter “Late arriving CPNs develop altered callosal projections”.

The lack of activity in ipsilateral hemisphere as measured by SAREdGFP expression at P10 was indeed in partial contradiction with the intracortical recording data, that does not discover any anomalies in this hemisphere. However, it should be noted, that electrophysiological recordings were performed at P21. Knowing that the sensitivity of the two methods are probably not comparable, we decided to verify activity at P21 also by quantification of SAREdGFP positive electroporated cells. As can be seen in the new Fig. 5a, we have not seen the previously observed difference between control and dnTCF4 electroporated hemispheres when we analyzed brains at P21, which let us speculate a merely transient downregulation of neuronal activity.

Contralateral activity was measured in order to test transcallosal connectivity after callosal development at P35. Enriched environment in this context was necessary to enhance the activity of the somatosensory pathway. However, when ipsilateral activity was assessed at P10 and P21, we wanted to evaluate whether electroporated cells could be activated in a “natural” developmental context. Therefore, we did not apply the enhanced activity paradigm.

13) Comment: « The manuscript and figures would benefit from editing to amend typographical errors and improve the clarity of writing. »

Response: We fully agree with this opinion. In the revised manuscript, we have payed attention to avoid typo and attempted to improve the clarity of our text.

We thank you again for the insightful comments; we hope that the outlined new experimental data together with our point-by-point response to the Referee’s comments are sufficiently convincing to consider our manuscript for publication in Nature Communication.

Below you may find a point-to-point response to the reviewers' comments:

Reviewer 1 and 4

We thank the Reviewer for the positive note, and for the constructive comments during the review process.

Reviewer 2

1) Comment: « The authors have done a nice job of responding to most of the concerns. »

Response: We thank the reviewer for the positive remark.

2) Comment: « I still believe it is difficult to know whether it is the migration "delay" that results in the phenotype or whether it is separate effects on activity caused by changes in Wnt signaling. The authors persist in making this overblown conclusion. They have modulated it in some places but it is still obvious that this is their bias without adequate proof (eg see the title).»

Response: We accept this criticism and we agree that this issue cannot be definitively clarified in this paper without using an alternative method to delay cell migration. Therefore, we have tuned down our conclusions throughout the text and have even changed the title of the manuscript.

3) Comment: « It seems that the authors really need to reproduce this [*i.e. C-Kit overexpression*] and determine whether it has the opposite effects of their Wnt manipulations. »

Response: We agree that this point could reinforce our interpretations of the data. Recently, we have carried out these experiments and we are able to add the requested new sets of data to the revised manuscript. It is important to emphasize that compared to Guijarro et al., we perform only a transitional overexpression by our inducible construct, limited to the period of migration (i.e. E21-P3). In the revised manuscript, we now show that C-kit overexpression speeds up radial migration in our hands, as evidenced by cell positioning at P3 (Supplementary Fig. 3i) and direct measurement of migratory speed by time-lapse video microscopy (Supplementary Fig. 3h). Thus, C-kit overexpression has opposing effects on radial migration as the perturbation of Wnt signalling, however, we cannot exclude any unrelated long-term effects of the transient upregulation of C-kit signalling (see response to comment 2).

Reviewer 3

1) Comment: « The authors have made a significant effort to address all my concerns from my first review. I think the results presented are interesting and would be of interest to the readers of Nature Communications. »

Response: We thank the reviewer for the positive comment, and for the constructive suggestions throughout the review process.

2) Comment: « The results showing that changes in neuronal migration affect callosal projections and mouse behavior were most extensively demonstrated with dnTCF. I understand that the authors provided substantial evidence that this is through the function of TCF in canonical Wnt signaling. However, in an overexpression system there are alternative explanations, and the results may be specific to a weak suppression of nuclear Wnt signaling without affecting cell adhesion at a specific developmental stage and in specific neurons. I would suggest a little more emphasis on dnTCF and TCF. For example, P5, 3 lines from bottom, change to “dnTCF did not seem to affect the radial glia structure.” This comment does not affect the novelty of the manuscript. »

Response: We thank the reviewer for the precise suggestion. We accept this suggestion and accordingly, we have modified the text in the revised manuscript.

3) Comment: « On P10, line 16, an accurate description is: “...that delayed neurons had reduced mEPSC frequency” or “the delayed neurons had longer inter-event intervals”. In the associated Fig. 5c, should be “mEPSC amplitude” and “mEPSC inter-event interval”. »

Response: We thank the reviewer for this correction. We agree and have modified the text in the revised manuscript.

Reviewer 2

1) Comment: « The authors have addressed my residual concerns. »

Response: We thank the reviewer for the positive comment and for the constructive suggestions throughout the review process.